# Spike-driven Large Language Model

## Abstract

Inspired by the low-energy characteristics of biological computing mechanisms, Spiking Neural Networks (SNNs), with their spike-driven operations and spatiotemporal dynamics, offer a promising solution for constructing energy-efficient language models. Although prior research has attempted to integrate SNNs with Large Language Models (LLMs), these approaches often suffer from limited performance or low inference efficiency. To tackle these challenges, we propose a Spike-driven Large Language Model (SDLLM) that enables large-scale modeling by eliminating matrix multiplications and relying solely on sparse additions. Specifically, we propose a two-step spike quantization strategy to address the numerous outliers in LLM activation values, significantly mitigating the accuracy loss caused by binary spike trains. To further reduce the spike firing rate, we introduce bidirectional encoding under symmetric quantization, along with a membrane potential clipping mechanism, which together reduce energy consumption without compromising accuracy. Extensive experiments demonstrate that SDLLM performs effectively on both language modeling and commonsense QA tasks. For example, compared to previous spike-based LLMs, our SDLLM reduces energy consumption by $7.8\times$ and improves accuracy in common scene reasoning by $4.2\%$. SDLLM is the first to demonstrate that SNNs outperform quantized artificial neural networks (ANNs) in both performance and energy efficiency, and can serve as a low-energy algorithmic approach to guide the collaborative design of neuromorphic hardware, exhibiting superior performance and energy efficiency in LLM scenarios.

## 1 Introduction

Large Language Models (LLMs) have emerged as a significant breakthrough in artificial intelligence research, gaining considerable attention for their exceptional performance (Touvron et al., 2023a;b; Zhang et al., 2022a) in natural language processing, knowledge reasoning, and generative tasks. However, the deployment of LLMs faces substantial computational and storage challenges, especially on resource-constrained devices (Shao et al., 2023). In contrast, the human brain efficiently performs complex tasks with a power consumption of less than 20 watts, posing a new challenge for the energy efficiency of AI systems (Balasubramaniana, 2021). Spiking Neural Networks (SNNs), inspired by the low-energy characteristics of biological computation, offer a promising approach for energy-efficient language modeling. Leveraging their unique spike-driven mechanism (Yao et al., 2024b) and spatiotemporal dynamics (Maass, 1997), SNNs present an opportunity to optimize energy consumption in language tasks. Therefore, there is an urgent need for low-bit and high-performance Spike-based LLMs.

Initially, SNNs were mainly used for visual tasks, where the optimization demands for spike representation, sparsity, and time steps are much lower than those of LLMs. As a result, SNNs perform well in visual tasks but are difficult to transfer directly to LLMs (Luo et al., 2024; Yao et al., 2025; Liu et al., 2025). Numerous efforts have been made to integrate SNNs with LLMs. However, simply combining SNNs with LLMs either results in insufficient performance or low inference efficiency. For instance, some works (Lv et al., 2023; Zhu et al., 2023; Xing et al., 2024b) combine SNNs with NLP models such as BERT or GPT (Devlin et al., 2019; Radford et al., 2021). However, their parameter scale remains limited to millions, making them suitable only for small supervised tasks and prone to performance degradation as data scale increases. SpikeLLM (Xing et al., 2024a) pushes generative tasks forward by converting a 7B-parameter Transformer into a spiking version, achieving promising results. Nevertheless, it relies on 8-bit high activation values to compensate for performance, which

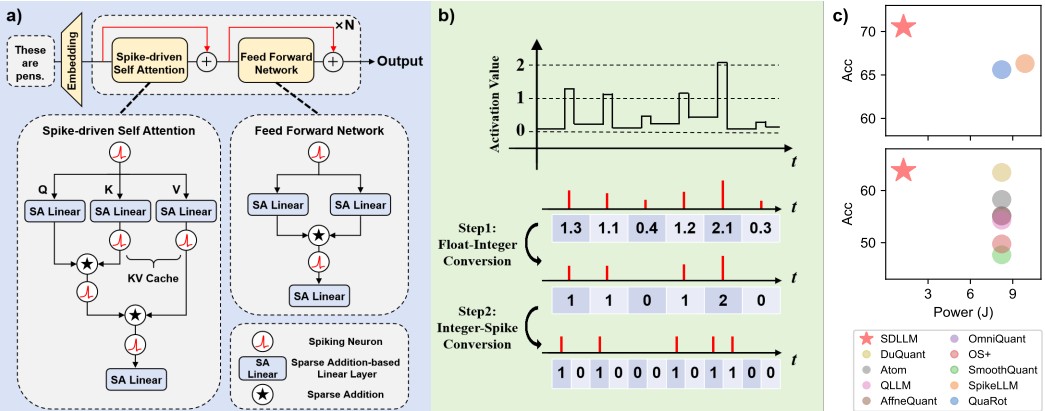

Figure 1: (a) The architecture of Spike-Driven LLM. (b) Two steps for quantizing spike neurons. (c) Performance and energy comparison of SDLLM vs. W4A4 SpikeLLM (top) and standard W4A4 quantization (bottom) on LLaMA-2 13B baseline.

undermines the spike-driven nature and hinders the exploitation of spike-based sparsity, as well as adaptation to neuromorphic hardware.

In this work, we aim to develop a spike-driven language model with large-scale capacity, balanced performance, and low power consumption, focusing on efficient sparsity utilization in SNNs. To address the performance gap between ANNs and SNNs, we replace conventional matrix multiplication with binary spike-based operations during inference, achieving energy-efficient sparse additions. Our two-step method constructs spike neurons by first quantizing continuous membrane potentials into integer spike counts, then expanding these counts into binary spike trains for event-driven computation. Controlling the spike firing rate is crucial, impacting computational load and energy consumption. Reducing higher spike count probabilities effectively lowers firing rates. Compared with quantization schemes, symmetric methods result in lower firing rates due to the reduced mapping region as spike counts increase. To enhance sparsity, we propose a ReLU-based variant that truncates the membrane potential distribution before quantization and combines it with rotation matrices to reduce quantization error. Our contributions are summarised as follows:

- We design and implement the first sparse addition-based spike-driven LLM We employ a two-step spike quantization method that significantly mitigates the accuracy loss caused by 0/1 spike encoding, achieving performance comparable to or even surpassing mainstream soft quantization approaches in ANN under equivalent bit-width.

- We significantly reduce the spike firing rate by incorporating two techniques: bidirectional encoding under symmetric quantization and membrane potential clipping. Under equivalent bit-width, our method achieves lower operations and up to $13\times$ reduction in energy consumption compared to ANN quantization methods, demonstrating the strong advantages of spike-based models over their ANN counterparts. At the same time, our method can provide guidance for the customization and optimization of low-energy neuromorphic hardware at the software-hardware co-design level.

## 2 RELATED WORKS

**Training of Spiking Neural Networks**    The development of SNNs has long been hindered by the challenge of training non-differentiable binary spikes. To address this, researchers have focused on improving training methods and architectural designs. Recently, two primary methods for training high-performance SNNs have emerged. One approach is to convert ANNs into spike form through neuron equivalence (Li et al., 2021; Hao et al., 2023), known as ANN-to-SNN conversion. However, this method requires long simulation time steps and increases energy consumption. We employ the direct training method (Wu et al., 2018) and apply surrogate gradient training.

**Spiking Neural Networks for Natural Language Processing**     As LLMs like GPT-3 scale, their rising computational and energy demands raise cost and sustainability concerns. To address this, SNNs are being explored in NLP for energy-efficient modeling. Bi-SNN (Xiao et al., 2022) introduced a bidirectional SNN for sentiment classification and translation. SpikingBERT (Lv et al., 2023; Bal & Sengupta, 2024) and SpikeLM (Xing et al., 2024b) combined SNNs with BERT via spike-based distillation and dual-spike encoding, but remain limited to million-scale parameters and small supervised tasks. SpikeGPT (Zhu et al., 2023) adopted binary spike activations and simplified attention to reduce computation, yet suffered from scaling issues. SpikeLLM (Xing et al., 2024a) scaled SNNs to 7 B-parameter Transformers using a "best-brain" framework, achieving competitive results, but relied on 8-bit high activations, weakening spike sparsity and neuromorphic compatibility.

**Model Compression**     Various compression techniques have been explored to reduce the scale of large SNNs, including: (i) Sparsification in SNNs (Han et al., 2015; Wei et al., 2025), which typically adapts pruning techniques from traditional ANNs to suit both the spatial and temporal domains of spiking models (Shi et al., 2023; Shen et al., 2024). While effective on simple datasets and shallow networks, achieving strong performance on complex tasks and deeper architectures remains challenging. (ii) Knowledge distillation (Hinton et al., 2015) transfers knowledge from large ANNs or SNNs into smaller SNNs to reduce model size and power consumption. However, numerous methods (Takuya et al., 2021; Xu et al., 2023a) only distill final output logits, leading to incomplete knowledge transfer and limited effectiveness in downstream SNN performance. (iii) Quantization (Jacob et al., 2018; Krishnamoorthi, 2018), especially relevant for hardware deployment, reduces bit-widths of weights and activations, enabling energy-efficient inference. Recent studies on SNN quantization (Deng et al., 2021; Qiu et al., 2025) have focused on Quantization-Aware Training (QAT) techniques for convolutional and transformer models in vision tasks, achieving strong performance with task-specific training protocols. However, such methods are not directly applicable to spike-based LLMs. Some recent efforts (Xing et al., 2024a; Shao et al., 2023; Liu et al., 2024b) have explored adapting post-training quantization (PTQ) to spike-based LLMs. In this work, we propose a method that directly maps quantized LLM activations to spike trains, maintaining the spike-driven nature of inference while enabling scalable and efficient deployment for spike-based LLMs.

## 3   PRELIMINARY

**Quantization Framework**     We employ uniform quantization for both weights and activations to enhance hardware compatibility and efficiency. For a full-precision matrix $\mathbf{X}$, the $N$-bit quantization process is as follows:

$$\hat{\mathbf{X}} = \text{Clamp}\left(\left\lfloor \frac{\mathbf{X}}{\alpha} \right\rceil + \mathbf{Z}, 0, 2^N - 1\right), \text{ where } \alpha = \frac{\text{Max}(\mathbf{X}) - \text{Min}(\mathbf{X})}{2^N - 1}, \mathbf{Z} = -\left\lfloor \frac{\text{Min}(\mathbf{X})}{\alpha} \right\rceil, \quad (1)$$

where $\hat{\mathbf{X}}$ is the quantized counterpart, $\alpha$ is the quantization step size, $\lfloor \cdot \rceil$ is the rounding function, and $Z$ represents the zero-point value. Moreover, $\text{clip}\{x, a, b\}$ confines $x$ within range $[a, b]$. The quantization process described above can be expressed using the quantization function $Q(\cdot)$.

**LIF Spike Neuron**     The Leaky Integrate-and-Fire (LIF) neuron is a simplified biologically inspired model that simulates the electrical activity of biological neurons (Roy et al., 2019). It integrates incoming signals while accounting for the gradual decay (leakage) of membrane potential over time. When the membrane potential reaches a threshold, a spike is generated and the potential is reset to a baseline. Due to its balance between computational simplicity, efficiency, and biological plausibility, the LIF model is widely used in neuroscience and computational models to simulate neural information processing. The update process is defined as follows:

$$\mathbf{v}^{(\ell)}[t] = \mathbf{h}^{(\ell)}[t-1] + f(\mathbf{w}^{(\ell)}, \mathbf{x}^{(\ell-1)}[t]), \quad (2)$$

$$\mathbf{s}^{(\ell)}[t] = \boldsymbol{\Theta}(\mathbf{v}^{(\ell)}[t] - \vartheta), \quad (3)$$

$$\mathbf{h}^{(\ell)}[t] = \mathbf{v}^{(\ell)}[t] \cdot (1 - \mathbf{s}^{(\ell)}[t]) + \mathbf{v}_{reset} \cdot \mathbf{s}^{(\ell)}[t]. \quad (4)$$

Here, the membrane potential $\mathbf{v}^{(\ell)}[t]$ at time step $t$ is updated based on the previous potential $\mathbf{h}^{(\ell)}[t-1]$ and the input signal $f(\mathbf{w}^{(\ell)}, \mathbf{x}^{(\ell-1)}[t])$, as shown in Eq. 2. A spike is triggered when the

potential exceeds the threshold $\vartheta$, with the step function $\Theta$ in Eq. 3 indicating the firing decision. If a spike occurs, the membrane potential is reset to $\mathbf{v}_{\text{reset}}$, as shown in Eq. 4.

# 4 METHOD

## 4.1 TWO STEPS FOR QUANTIZING SPIKE NEURONS

We aim to construct a sparse computation-based LLM driven by spikes. In the inference phase, we replace traditional matrix multiplication operations with 0/1 spike operations, thereby achieving more energy-efficient model computation through sparse addition. However, during the replacement from traditional ANN to SNN, performance degradation due to quantization becomes a significant challenge. To effectively reduce spike quantization errors, we propose a two-step quantization method to optimize the performance of spike neurons. As illustrated in Fig. 1(b), we first quantize the continuous membrane potential into integer-form spike counts; then, through time-domain expansion, we further map these integer spikes into 0/1 spike trains, enabling event-driven discrete computation.

**Step One: Integer-LIF Spike Neuron** The Integer Leaky Integrate-and-Fire (I-LIF) neuron is designed to reduce quantization errors in SNNs (Luo et al., 2024), improving performance in low-power scenarios. Unlike traditional SNNs that convert membrane potentials directly into binary spikes, which often causes representational loss. I-LIF uses integer-valued activations to enhance stability and training efficiency. For its dynamic process, we rewrite Eq. 3 as:

$$\mathbf{s}^{(\ell)}[t] = \text{Clip}(\text{Round}(\mathbf{v}^{(\ell)}[t]), 0, D). \tag{5}$$

At each time step $t$, the spike signal $\mathbf{s}^{(\ell)}[t]$ is generated by rounding and clipping the membrane potential $\mathbf{v}^{(\ell)}[t]$, ensuring that the spike value lies within the range $[0, D]$. We use the resulting integer spike as the spike count for the second step, where it is expanded into a 0/1 spike train.

**Step Two: From Integer Spike to 0/1 Spike** Spike counts in integer form are converted to traditional 0/1 spike values by extending the virtual time step from $T$ to $T \times D$ (Luo et al., 2024). Specifically, the input $\mathbf{s}^{(\ell)}[t]$ is extended into a spike train $\{\mathbf{s}^{(\ell)}[t, d]\}_d^D$, effectively converting integer values into traditional spike values, performing computations without matrix multiplication. The corresponding equations are given as follows:

$$\mathbf{v}^{(\ell+1)}[t] = \mathbf{h}^{(\ell+1)}[t-1] + \sum_d^D \left( \mathbf{w}^{(\ell+1)} \mathbf{s}^{(\ell)}[t, d] \right). \tag{6}$$

Since $\mathbf{w}^{(\ell+1)} \sum_d^D \mathbf{s}^{(\ell)}[t, d] = \sum_d^D \left( \mathbf{w}^{(\ell+1)} \mathbf{s}^{(\ell)}[t, d] \right)$, where $\mathbf{w}^{(\ell+1)}$ is the corresponding weight matrix, the spike $s^{(\ell)}[t, d]$ can thus replace matrix multiplication with sparse addition, *(Appendix C)*.

We design a Sparse Addition-based Linear Layer based on spike neurons, and construct a spike-driven LLM without matrix multiplication, relying solely on sparse addition operations, based on the LLaMA architecture (Fig.1(a)). It is worth noting that the majority of computation in large models is concentrated in matrix multiplication operators, while other operators, including bias, typically contribute several orders of magnitude less computational cost. Furthermore, RMSNorm has been empirically shown to be efficiently implementable on neuromorphic hardware using sparse addition operations (Abreu et al., 2025), and other nonlinear operators, such as GELU, Softmax, and the natural exponential function, can be approximated using Taylor series expansion (Arora et al., 2024).

## 4.2 ANALYSIS AND CHALLENGES OF SPIKE FIRING

The event-driven mechanism makes the firing rate a key factor in determining computational energy consumption. During the spike-based quantization process, the network exhibits inherent sparsity by quantizing floating-point values into spike counts, which are further expanded into 0/1 spike trains with specific firing rates. Based on this, we further investigate the regularity of sparsity in the quantized spike representation.

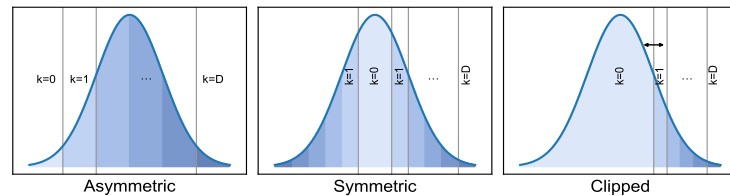

Figure 2: Different methods of spike quantization methods. Clipped method has adjustable 0-1 boundary, the other thresholds are uniformly split among 0 and saturation value D.

**Statistic spike firing count**   We first investigate the spiking sparsity resulting from different integer values obtained in the first step of our quantization method, as defined in Eq. 5. Since these integer values correspond to the number of spike fired of binary spike trains within the time window during inference, we denote this as k:

$$k^{(\ell)} = \sum_t^T \sum_d^D \mathbf{s}^{(\ell)}[t, d]. \tag{7}$$

$k^{(\ell)}$ denotes the total number of spike fired by the $\ell$-th layer neuron over the time window. $\mathbf{s}^{(\ell)}[t, d]$ is the spike state at time step $t$ and virtual step $d$ (1 if a spike is fired, 0 otherwise). The total count $k^{(\ell)}$ is obtained by summing $\mathbf{s}^{(\ell)}[t, d]$ over all $t$ and $d$.

**Calculate Spike Firing Rate**   As shown in the left panel of Fig. 2, for integer spike counts $0, 1, \ldots, T$ (defined in Eq. 7), the probability of each integer corresponds to the area of a specific interval under the membrane potential probability density function. The density function is divided into $T$ intervals, with each corresponding to one integer value. The area under each interval represents the probability of that value, denoted as $P$. The firing rate is expressed as follows:

$$R^{(\ell)} = \sum_k \frac{k^{(\ell)}}{T} \cdot P_k^{(\ell)}. \tag{8}$$

In this formula, $R^{(\ell)}$ denotes the firing rate of the $\ell$-th layer neuron, where $k^{(\ell)}$ is the integer spike count, $T$ is the time window length, and $P_k^{(\ell)}$ is the probability of quantizing to integer $k$. The firing rate is obtained by a weighted sum over all integer spike counts and their corresponding probabilities.

Since time steps are skipped when no spikes are fired, real $T$ is defined as $\boldsymbol{T} := \boldsymbol{T_D} \times \boldsymbol{R}$, where $T_D$ represents the extended time steps and $R$ is the firing rate. We visualize the spike firing counts across different layers of LLaMA2-7B in Fig. 3, where the first-step quantization adopts W4A4 ($T = 1$) and the second-step quantization adopts W4A1 ($T = 7.5$). Taking the QKV layer as an example, the input spike count reaches 7.52 with a firing rate of 0.5, indicating a clear spike redundancy.

### 4.3 More Sparsity Achieved under Symmetric Spike Quantization

To reduce the spike firing rate, we begin by analyzing the inherent sparsity patterns in the spike quantization process and the relationship between membrane potential and spike firing probability.

**Theorem 1** (The Relationship Between $R^{(\ell)}$ and $P_k$). *To reduce the spike firing rate $R^{(\ell)}$, smaller integer spike counts $k$ should correspond to larger probabilities $P_k$.*

*Proof.* From equation (8), the spike firing rate $R^{(\ell)}$ is given by: $R^{(\ell)} = \sum_k \frac{k^{(\ell)}}{T} \cdot P_k^{(\ell)}$. Let $k_1 < k_2 < \cdots < k_n$ denote the integer spike counts, with corresponding quantization probabilities $P_{k_1} > P_{k_2} > \cdots > P_{k_n}$. When $k$ decreases, $P_k$ increases. Since $k_1$ is the smallest spike count, its corresponding probability $P_{k_1}$ is the largest, and its contribution to the overall firing rate is dominant. To minimize the firing rate $R^{(\ell)}$, smaller $k$ should be paired with larger $P_k$. This allocation minimizes $R^{(\ell)}$, producing a sparser binary spike train within a unit time window. □

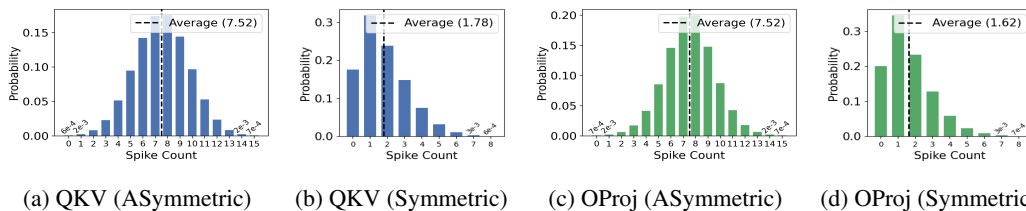

(a) QKV (ASymmetric)  (b) QKV (Symmetric)  (c) OProj (ASymmetric)  (d) OProj (Symmetric)

Figure 3: Significant reduction in spike count after symmetric quantization and bidirectional encoding. *More results can be found in Appendix B.*

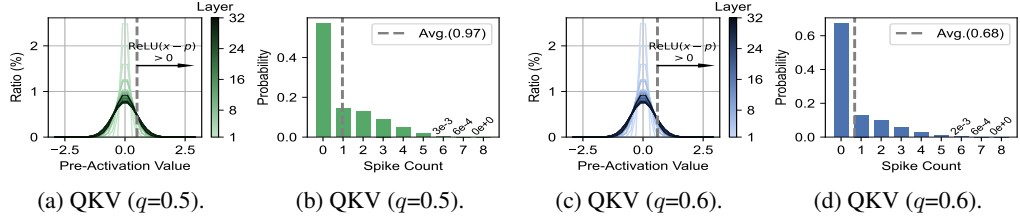

(a) QKV ($q$=0.5).  (b) QKV ($q$=0.5).  (c) QKV ($q$=0.6).  (d) QKV ($q$=0.6).

Figure 4: Spike count is further reduced by membrane potential clipping via quantile-based ReLU.

As illustrated in the left panel of Fig. 2, the previously adopted asymmetric quantization tends to concentrate the spike probability around middle integer values, with lower probabilities at both ends. To address this, we introduce a new encoding scheme using ternary spikes $-1/0/1$ to enable bidirectional encoding, allowing the membrane potential to be discretized through symmetric quantization (*see Appendix A for details and formulas on two-step spike quantization and spike firing count under bidirectional encoding*). In this approach, the membrane potential is mapped to a symmetric integer range $k \in \left(-\frac{D}{2} - 1, \frac{D}{2}\right)$, where positive and negative $k$ represent spike counts in the positive and negative directions, respectively. The extended time step is halved to $\frac{D}{2}$.

As shown in the middle panel of Fig. 2, in the symmetric spike quantization mode, the mapping range of the membrane potential narrows as the number of spikes within the unit time window increases, causing the mapping probability to decrease, resulting in a significantly lower overall firing rate compared to the asymmetric mode. In Fig. 3, we present the results on the QKV layer of LLaMA, where the average spike count is reduced from 7.52 to 1.78 and the firing rate decreases from 0.5 to 0.22 under symmetric quantization.

### 4.4 MORE SPARSITY ACHIEVED VIA MEMBRANE POTENTIAL CLIPPING

In addition to symmetric spike quantization, we further explore enhancing sparsity by modifying the initialization of the membrane potential distribution through clipping. As shown in Fig. 2 (right), the majority of the membrane potential distribution is mapped to the spike count of 0, while only a small clipped portion of the distribution is progressively mapped to spike counts from 1 to the maximum value. This design significantly increases the proportion of the probability mapping area corresponding to zero spikes, thereby further reducing the overall spike firing rate (see Fig. 4).

**Quantile-Shifted ReLU**   The ReLU (Rectified Linear Unit) activation function has been validated for its sparsity in traditional models. Inspired by this, we propose a variant more suitable for the problem in this paper— the quantile-shifted rectified unit activation function, which is applied to the train-level sparsity problem to address the challenges in membrane potential correction. We define the Quantile-Shifted Rectified Linear Unit Activation Function as:

$$\mathbf{v}_{sp}^{(\ell)}(t) = \text{ReLU}\left(\mathbf{v}^{(\ell)}(t) - \text{Quantile}(\mathbf{v}^{(\ell)}(t), q)\right). \tag{9}$$

$\mathbf{v}^{(\ell)}(t)$ represents the membrane potential of the $\ell$-th layer neuron at time step $t$, $q$ is the quantile ratio, and $\text{Quantile}(\mathbf{v}^{(\ell)}(t), q)$ calculates the membrane potential corresponding to the $q$-th quantile. The final processed value, $\mathbf{v}_{sp}^{(\ell)}(t)$, is obtained by applying the ReLU function and shifting the quantile

threshold. The ReLU function, based on the $q$-th quantile threshold, retains only the values above this threshold, which are used for sparsification. Then, We rewrite Eq.5 as:

$$\mathbf{s}^{(\ell)}[t] = \text{Clip}(\text{Round}(\mathbf{v}_{\text{sp}}^{(\ell)}(t)), 0, D). \tag{10}$$

**Joint Sparsity and Rotation Matrices**   Rotation matrices help reduce quantization information loss by uniformizing data distributions. We incorporate them into our spike-based quantization framework and explore their combination with sparsification. By learning sparsity from computational invariance, we utilize $XW = XQQ^TW$, where $Q$ is an orthogonal rotation matrix (Ashkboos et al., 2024), and construct $\text{ReLU}(XQ)Q^TW$ to enhance quantization performance and spike sparsity. Eq.9 becomes:

$$\mathbf{v}_{\text{sp}}^{(\ell)}(t) = \text{ReLU}\left(\mathbf{v}^{(\ell)}(t)Q - \text{Quantile}(\mathbf{v}^{(\ell)}(t)Q, q)\right). \tag{11}$$

For the next layer, we have: $\mathbf{v}^{(\ell+1)}[t] = \mathbf{h}^{(\ell+1)}[t-1] + f(\mathbf{w}^{(\ell+1)}Q, \mathbf{s}^{(\ell)}[t])$. In Eq. 11, the membrane potential is transformed by rotation matrix $Q$ and adjusted by the quantile ReLU to generate sparse potentials. These are quantized into spike signals, which are then multiplied by rotated weights $\mathbf{w}^{(\ell+1)}Q$, enabling sparse addition-based updates to the next-layer membrane potential.

# 5 EXPERIMENTS

**Models and Evaluations**   We apply our spike-driven approach to the LLaMA family of pre-trained LLMs (and the newer LLM Qwen2.5 (Team, 2024)) and systematically evaluate the performance on commonsense question answering (PIQA (Bisk et al., 2020), ARC-easy (Clark et al., 2018), ARC-challenge (Clark et al., 2018), HellaSwag (Clark et al., 2018), and WinoGrande (Sakaguchi et al., 2021)) and more complex language generation, including reading comprehension (BoolQ (Clark et al., 2019), SQuAD (Rajpurkar et al., 2016)), world knowledge (TriviaQA (Touvron et al., 2023a)), and math (GSM8K (Cobbe et al., 2021)).

**Implementation Details**   (i) Our evaluation focuses on 4-bit quantization, with a value range of $0 \sim 2^4 + Z_0$, representing 16 integer values, where $Z_0$ is the zero-point used to offset the quantization. To clearly compare SNNs and ANNs, we provide the corresponding quantization ranges in the results. For example, in $A1.5T_D8$, $A1.5$ represents symmetric encoding, and multiplying by $T_D8$ is used to supplement the quantization values, i.e., $A1.5T_D8 = \{-8, -7, \dots, -1\} \cup \{0, 1, \dots, 7\}$, with $2 \times 8$ values in total. (ii) We performed offline quantization on all inputs, weights, and KV caches using online quantization, without the need for training any quantization parameters. We adapted the rotation matrix method from the QuaRot paper (Ashkboos et al., 2024) and further optimized it. (iii) The sparse rotation training configuration and additional details are provided in Appendix D.

**Operations and Energy Consumption**   (i) As in the SpikeLLM paper, we adopt the ACE metric (Xing et al., 2024a; Zhang et al., 2022b) to measure the total number of binary operations in the model, $\text{ACE} = \text{MACs} \times \text{bit}_{\text{weight}} \times \text{bit}_{\text{act.}}$. (ii) As in the previous paper, due to the different computational overhead of quantized values compared to floating-point values (Wang et al., 2020), we use 1/32 FLOPs to represent 2-bit operations (Xu et al., 2023b; Liu et al., 2020). Similarly, for 4-bit $\times$ 4-bit, it's equivalent to $4\times$ (2-bit $\times$ 2-bit). (iii) Regarding power consumption, due to the different energy costs between a single multiplication and a single addition, we follow previous standards (Yao et al., 2025; 2024a) to estimate power. (iiii) For ease of comparison between ANN and SNN, we set $T$ for ANN to 1, and and calculate the above metrics $\times T$. (*See Appendix F for details.*)

## 5.1 MAIN RESULTS

**Comparisons with SpikeLLM**   As shown in Tab. 1, we compare SDLLM (W4A1.5, $T = 1.73$) with SpikeLLM (W4A4, $T = 1.2$), where both models are improved based on QuaRot rotation-based quantization and use RTN for weight quantization. Since A1.5 $\times$ T8 $<$ A4 $\times$ T1.2, SpikeLLM actually uses a higher number of activation bits than ours. Experimental results demonstrate that SDLLM achieves performance improvements of 5.69% and 4.23% over SpikeLLM on the LLaMA-2-7B and LLaMA-2-13B models, respectively. Moreover, compared to SpikeLLM, SDLLM reduces ACEs by $1.39\times$, FLOPs by $1.39\times$ and energy consumption by $7.58\times$.

Table 1: Zero-shot QA (↑) results between SDLLM and SpikeLLM under SpikeLLM settings.

| Method | Spike | W | A | | | PIQA | ARC-e | ARC-c | BoolQ | HellaS | WinoG | Avg. | ACEs | Flops (T) | Power (J) |
|---|---|---|---|---|---|---|---|---|---|---|---|---|---|---|---|
| | | Bit | Bit | $T_{=T_D\times R}$ | $Range_{(+Z_0)}$ | | | | | | | | | | |
| LLaMA-2-7B | ✗ | - | - | - | - | 78.84 | 74.54 | 46.33 | 77.74 | 75.97 | 69.22 | 70.44 | 1× | 6.91 | 33.84 |
| QuaRot | ✗ | 4 | 4 | - | $0\sim2^4$ | 71.82 | 59.89 | 36.18 | 67.37 | 63.88 | 59.12 | 59.71 | 0.063× | 0.86 | 4.23 |
| SpikeLLM | ✓ | 4 | 4 | 1.2 | $0\sim2^4$ | 72.47 | 62.29 | 36.01 | 69.48 | 64.74 | 59.43 | 60.74 | 0.075× | 1.04 | 5.08 |
| SDLLM | ✓ | 4 | 1.5 | $1.73_{=8\times0.216}$ | $0\sim2^4$ | 75.84 | 69.65 | 41.21 | 74.01 | 71.75 | 66.14 | **66.43** | 0.054× | **0.75** | **0.67** |
| LLaMA-2-13B | ✗ | - | - | - | - | 80.63 | 77.48 | 49.23 | 80.73 | 79.37 | 71.74 | 80.69 | 1× | 13.42 | 65.77 |
| QuaRot | ✗ | 4 | 4 | - | $0\sim2^4$ | 74.86 | 69.19 | 41.98 | 72.54 | 70.35 | 64.72 | 65.61 | 0.063× | 1.68 | 8.22 |
| SpikeLLM | ✓ | 4 | 4 | 1.2 | $0\sim2^4$ | 75.79 | 69.53 | 41.21 | 74.31 | 71.51 | 65.51 | 66.31 | 0.075× | 2.01 | 9.87 |
| SDLLM | ✓ | 4 | 1.5 | $1.67_{=8\times0.209}$ | $0\sim2^4$ | 78.51 | 74.12 | 46.16 | 78.26 | 76.36 | 69.85 | **70.54** | 0.052× | **1.40** | **1.26** |

Table 2: Zero-shot QA (↑) with Membrane Potential Clipping: Lower Firing Rate Enhances Efficiency.

| Method | QKV | | | | | | PIQA | ARC-E | ARC-C | BoolQ | HellaS | WinoG | Avg. |
|---|---|---|---|---|---|---|---|---|---|---|---|---|---|
| | $q$ | $T_{=T_D\times R}$ | $Range_{(+Z_0)}$ | ACEs | FLOPs(T) | Power(J) | | | | | | | |
| LLaMA-2-7B | - | - | - | 1× | 1.649 | 8.081 | 78.84 | 74.54 | 46.33 | 77.74 | 75.97 | 69.22 | 70.44 |
| QuaRot-W4A4 | - | - | $0\sim2^4$ | 0.063× | 0.206 | 1.010 | 71.82 | 59.89 | 36.18 | 67.37 | 63.88 | 59.12 | 59.71 |
| SpikeLLM-W4A4 | - | 1.2 | $0\sim2^4$ | 0.075× | 0.247 | 1.212 | 72.47 | 62.29 | 36.01 | 69.48 | 64.74 | 59.43 | 60.74 |
| SDLLM-W4A1.5 | - | $1.73_{=8\times0.216}$ | $0\sim2^4$ | **0.054×** | **0.184** | **0.166** | 75.84 | 69.65 | 41.21 | 74.01 | 71.75 | 66.14 | **66.43** |
| SDLLM-W4A1.5 | 0.5 | $0.96_{=8\times0.120}$ | $0\sim2^4$ | **0.030×** | **0.100** | **0.090** | 73.94 | 59.22 | 34.30 | 71.71 | 64.30 | 63.61 | **61.18** |
| SDLLM-W4A1.5 | 0.6 | $0.80_{=8\times0.100}$ | $0\sim2^4$ | **0.025×** | **0.083** | **0.075** | 73.07 | 61.15 | 34.13 | 69.60 | 63.57 | 60.85 | 60.40 |

**Membrane Potential Clipping**  We evaluate the performance of spike-based models under the membrane potential clipping scheme, as shown in Tab. 2. Compared to SDLLM with symmetric quantization, applying a clipping threshold at the 60% quantile ($q = 0.6$) reduces the spike firing rate in the QKV layer from 0.22 to 0.10, leading to to a 2.2× reduction in ACEs, FLOPs and energy consumption. Compared to SpikeLLM, the QKV layer of SDLLM reduces energy consumption by 16× while maintaining comparable accuracy.

**Comparison with General Quantization**  We compare SDLLM with general quantization methods, such as SmoothQuant (Xiao et al., 2023), OS + (Wei et al., 2023), OmniQuant (Shao et al., 2023), AffineQuant (Ma et al., 2024), QLLM (Liu et al., 2024a), Atom (Zhao et al., 2023), DuQuant (Lin et al., 2024). As shown in Tab. 3 and 4, SDLLM outperforms DuQuant in Zero-shot QA tasks, achieving state-of-the-art (SOTA) performance on LLaMA-2-7B, LLaMA-2-13B, and LLaMA3-8B, while reducing ACEs by 1.17×, 1.21×, and 1.19×, FLOPs by 1.14×, 1.20×, and 1.19×, and energy consumption by 6.31×, 6.52×, and 6.51×, respectively. These results demonstrate the advantages of our proposed SDLLM. It not only establishes new SOTA performance in comparison with ANN quantization methods but also significantly reduces operations and energy consumption through sparse addition enabled by spike-driven computation. Additionally, we report SDLLM's strong results on Qwen2.5-14B and more complex tasks *in the Appendix F*.

## 5.2   ABLATION STUDY

**Ablation Results**   (i) **Improved Performance.** Increasing the time step $T$ from 1.73 to 1.79 in the W4A1.5 configuration boosts LLAMA-2-7B accuracy to 68.80%, reducing energy consumption by 3×, balancing performance and efficiency. (ii) **A1 vs A1.5.** A1.5 significantly reduces spike firing rate and time step in bidirectional encoding, cutting operations and energy by 4× (accuracy unaffected by encoding). (iii) **W6A6.** W6A6 increases model capacity but reduces SDLLM's energy efficiency. Compared to ANN quantization, W6A6 consumes 2× less energy, demonstrating high efficiency at higher bit-widths. (Tab. 5)

**Hardware Potential**   (i) **Spike Delay.** The real-time steps are very short, typically $T < 2$, and our algorithm adapts to various hardware architectures (serial, parallel, and parallel-reuse), maximizing hardware efficiency (*Appendix H*) (ii) **Ternary no-Matrix Multiplication Feasibility.** Previous work on Loihi 2 demonstrated ternary no-matrix multiplication's feasibility and energy efficiency advantages (Zhu et al., 2024). However, ternary weights with no-matrix multiplication cannot leverage sparse event-driven computation. In contrast, our ternary spikes enable sparser additions, significantly reducing computation and energy consumption (e.g., with a firing rate of 0.2, only 20% of neurons are active). (iii) **Inspiring Hardware Design.** These findings highlight the importance of

Table 3: Evaluation of Zero-shot QA (↑) results of LLaMA2-7B and 13B under QLLM settings.

| Method | Spike | W Bit | A Bit | $T_{=T_D \times R}$ | $Range_{(+Z_0)}$ | PIQA | ARC-e | ARC-c | BoolQ | HellaS | WinoG | Avg. | ACEs | Flops (T) | Power (J) |
|---|---|---|---|---|---|---|---|---|---|---|---|---|---|---|---|
| LLaMA-2-7B | ✗ | - | - | - | - | 76.88 | 53.54 | 40.53 | 71.13 | 72.96 | 67.25 | 63.72 | 1× | 6.91 | 33.84 |
| SmoothQuant | ✗ | 4 | 4 | - | $0\sim2^4$ | 60.17 | 35.23 | 27.13 | 57.92 | 37.08 | 49.57 | 44.52 | 0.063× | 0.86 | 4.23 |
| OS+ | ✗ | 4 | 4 | - | $0\sim2^4$ | 63.11 | 39.10 | 28.84 | - | 51.30 | 45.93 | 45.66 | 0.063× | 0.86 | 4.23 |
| OmniQuant | ✗ | 4 | 4 | - | $0\sim2^4$ | 65.61 | 44.28 | 30.38 | 62.66 | 53.51 | 51.85 | 51.38 | 0.063× | 0.86 | 4.23 |
| AffineQuant | ✗ | 4 | 4 | - | $0\sim2^4$ | 67.36 | 44.23 | 31.91 | 62.75 | 54.34 | 55.18 | 52.64 | 0.063× | 0.86 | 4.23 |
| QLLM | ✗ | 4 | 4 | - | $0\sim2^4$ | 67.68 | 45.29 | 32.09 | 62.42 | 58.45 | 56.59 | 51.60 | 0.063× | 0.86 | 4.23 |
| Atom | ✗ | 4 | 4 | - | $0\sim2^4$ | 69.75 | 47.35 | 34.22 | 62.42 | 63.21 | 56.51 | 55.58 | 0.063× | 0.86 | 4.23 |
| DuQuant | ✗ | 4 | 4 | - | $0\sim2^4$ | 75.24 | 51.89 | 36.77 | 67.86 | 69.54 | 62.12 | 60.57 | 0.063× | 0.86 | 4.23 |
| SDLLM | ✓ | 4 | 1.5 | $1.73_{=8\times0.216}$ | $0\sim2^4$ | 74.54 | 51.89 | 38.74 | 68.81 | 69.00 | 63.54 | **61.09** | **0.054×** | **0.75** | **0.67** |
| LLaMA-2-13B | ✗ | - | - | - | - | 79.05 | 57.91 | 44.20 | 69.02 | 76.60 | 69.69 | 66.08 | 1× | 13.42 | 65.77 |
| SmoothQuant | ✗ | 4 | 4 | - | $0\sim2^4$ | 62.30 | 40.28 | 30.72 | 60.49 | 42.24 | 49.96 | 47.67 | 0.063× | 1.68 | 8.22 |
| OS+ | ✗ | 4 | 4 | - | $0\sim2^4$ | 64.47 | 41.46 | 32.17 | - | 59.30 | 51.38 | 49.76 | 0.063× | 1.68 | 8.22 |
| OmniQuant | ✗ | 4 | 4 | - | $0\sim2^4$ | 69.80 | 47.22 | 33.79 | 65.47 | 59.34 | 55.49 | 55.19 | 0.063× | 1.68 | 8.22 |
| AffineQuant | ✗ | 4 | 4 | - | $0\sim2^4$ | 68.55 | 47.64 | 32.34 | 66.97 | 59.97 | 55.07 | 55.09 | 0.063× | 1.68 | 8.22 |
| QLLM | ✗ | 4 | 4 | - | $0\sim2^4$ | 70.46 | 48.48 | 34.39 | - | 62.80 | 55.41 | 54.31 | 0.063× | 1.68 | 8.22 |
| Atom | ✗ | 4 | 4 | - | $0\sim2^4$ | 71.16 | 50.89 | 37.88 | 63.91 | 67.51 | 58.40 | 58.29 | 0.063× | 1.68 | 8.22 |
| DuQuant | ✗ | 4 | 4 | - | $0\sim2^4$ | 77.31 | 55.60 | 41.55 | 66.61 | 73.68 | 66.06 | 63.47 | 0.063× | 1.68 | 8.22 |
| SDLLM | ✓ | 4 | 1.5 | $1.67_{=8\times0.209}$ | $0\sim2^4$ | 77.26 | 57.41 | 41.55 | 66.67 | 73.33 | 66.69 | **63.82** | **0.052×** | **1.40** | **1.26** |

Table 4: Evaluation of Zero-shot QA (↑) results of LLaMA3-8B under DuQuant settings.

| Method | Spike | W Bit | A Bit | $T_{=T_D \times R}$ | $Range_{(+Z_0)}$ | PIQA | ARC-e | ARC-c | BoolQ | HellaS | WinoG | Avg. | ACEs | Flops (T) | Power (J) |
|---|---|---|---|---|---|---|---|---|---|---|---|---|---|---|---|
| LLaMA3-8B | ✗ | - | - | - | - | 80.85 | 77.78 | 53.41 | 81.28 | 79.16 | 72.84 | 74.22 | 1× | 7.97 | 39.06 |
| SmoothQuant | ✗ | 4 | 4 | - | $0\sim2^4$ | 54.57 | 31.9 | 24.23 | 52.72 | 31.26 | 51.14 | 40.97 | 0.063× | 1.00 | 4.88 |
| OmniQuant | ✗ | 4 | 4 | - | $0\sim2^4$ | 50.22 | 26.94 | 24.57 | 37.98 | 26.55 | 50.20 | 36.08 | 0.063× | 1.00 | 4.88 |
| AffineQuant | ✗ | 4 | 4 | - | $0\sim2^4$ | 50.71 | 25.93 | 26.02 | 40.55 | 26.07 | 48.46 | 36.29 | 0.063× | 1.00 | 4.88 |
| Atom | ✗ | 4 | 4 | - | $0\sim2^4$ | 62.95 | 49.45 | 30.12 | 60.31 | 53.75 | 56.04 | 52.10 | 0.063× | 1.00 | 4.88 |
| DuQuant | ✗ | 4 | 4 | - | $0\sim2^4$ | 75.68 | 68.48 | 41.81 | 71.99 | 73.07 | 66.22 | 66.21 | 0.063× | 1.00 | 4.88 |
| SDLLM | ✓ | 4 | 1.5 | $1.68_{=8\times0.210}$ | $0\sim2^4$ | 75.90 | 67.05 | 44.37 | 72.45 | 73.26 | 67.01 | **66.67** | **0.053×** | **0.84** | **0.75** |

Table 5: Ablation study of SDLLM for LLaMA2-7B (*13B in the Appendix Tab. S3*).

| Method | Spike | W Bit | A Bit | $T_{=T_D \times R}$ | $Range_{(+Z_0)}$ | PIQA | ARC-e | ARC-c | BoolQ | HellaS | WinoG | Avg. | ACEs | Flops (T) | Power (J) |
|---|---|---|---|---|---|---|---|---|---|---|---|---|---|---|---|
| LLaMA-2-7B | ✗ | - | - | - | - | 78.84 | 74.54 | 46.33 | 77.74 | 75.97 | 69.22 | 70.44 | 1× | 6.91 | 33.84 |
| SDLLM | ✓ | 4 | 1.5 | $1.73_{=8\times0.216}$ | $0\sim2^4$ | 75.84 | 69.65 | 41.21 | 74.01 | 71.75 | 66.14 | 66.43 | 0.054× | 0.75 | 0.67 |
| SDLLM | ✓ | 4 | 1.5 | $1.79_{=8.3\times0.216}$ | $0\sim2^4$ 92% $0\sim2^5$ 8% | 77.31 | 70.29 | 41.13 | 72.42 | 73.05 | 67.64 | 66.97 | 0.056× | 0.77 | 0.70 |
| SDLLM | ✓ | 4 | 1.5 | $3.54_{=16\times0.221}$ | $0\sim2^5$ | 78.02 | 72.05 | 44.28 | 75.87 | 74.49 | 68.11 | 68.80 | 0.111× | 1.53 | 1.37 |
| SDLLM$_{step1}$ | ✓ | 4 | 4 | 1 | $0\sim2^4$ | 75.84 | 69.65 | 41.21 | 74.01 | 71.75 | 66.14 | 66.43 | 0.063× | 0.86 | 4.23 |
| SDLLM | ✓ | 4 | 1.5 | $1.73_{=8\times0.216}$ | $0\sim2^4$ | 75.84 | 69.65 | 41.21 | 74.01 | 71.75 | 66.14 | 66.43 | 0.054× | 0.75 | 0.67 |
| SDLLM | ✓ | 4 | 1 | $7.5_{=15\times0.500}$ | $0\sim2^4$ | 75.84 | 69.65 | 41.21 | 74.01 | 71.75 | 66.14 | 66.43 | 0.117× | 3.24 | 2.92 |
| SDLLM$_{step1}$ | ✓ | 6 | 6 | 1 | $0\sim2^6$ | 78.89 | 74.58 | 45.56 | 76.57 | 75.80 | 68.98 | 70.06 | 0.141× | 1.94 | 9.52 |
| SDLLM | ✓ | 6 | 1.5 | $7.1_{=32\times0.222}$ | $0\sim2^6$ | 78.89 | 74.58 | 45.56 | 76.57 | 75.80 | 68.98 | 70.06 | 0.333× | 4.60 | 4.14 |
| SDLLM | ✓ | 6 | 1 | $31.5_{=63\times0.500}$ | $0\sim2^6$ | 78.89 | 74.58 | 45.56 | 76.57 | 75.80 | 68.98 | 70.06 | 0.738× | 20.41 | 18.37 |

algorithm-driven hardware design, offering insights for neuromorphic chip development and future hardware optimization.

# 6 CONCLUSION

In this work, we present the first spike-driven LLM that eliminates matrix multiplication entirely by leveraging sparse addition, built upon the LLaMA architecture. Unlike prior studies that only compared SNNs with full-precision ANNs, we are the first to systematically benchmark SNNs against mainstream ANN quantization methods. Our results demonstrate that, under equivalent bit-width settings, SDLLM achieves competitive accuracy while reducing energy consumption by up to 13×. This work provides the first compelling evidence that SNNs are not only feasible for large-scale models, but also possess the potential to rival quantized ANNs in both accuracy and energy efficiency, laying a critical foundation for the next generation of neuromorphic general intelligence.

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

## LIMITATIONS

Although our proposed Spike-driven Large Language Model (SDLLM) has made significant progress in both performance and energy efficiency, there are still some limitations to be addressed. Our model is based on sparse addition, and its inherent event-driven computation model requires hardware-level support, particularly for event-driven computation and sparse ternary spike representations. This support is crucial for fully realizing the potential of sparse addition and spike-based execution in large-scale implementations.

While existing research has optimized ternary weights without matrix multiplication on neuromorphic chips, demonstrating the feasibility of ternary values without matrix multiplication, it has not yet addressed the further need for optimization in event-driven computation and sparse capabilities. This work also highlights the importance of algorithm-guided hardware design. Our SDLLM algorithm provides significant insights for the next generation of neuromorphic hardware, contributing to the collaborative development of efficient neural chips that combine algorithmic advances with hardware optimizations.

## APPENDIX

## A  BIDIRECTIONAL ENCODING UNDER SYMMETRIC QUANTIZATION

### A.1  TWO STEPS SPIKE QUANTIZATION

**Step One**  As mentioned in Section 4.3, to address the high firing rate caused by asymmetrically quantized spikes, we adopt bidirectional spike encoding under symmetric quantization. We rewrite the dynamic process of the I-LIF neuron as:

$$\mathbf{v}^{(\ell)}[t] = \mathbf{h}^{(\ell)}[t-1] + f(\mathbf{w}^{(\ell)}, \mathbf{x}^{(\ell-1)}[t]), \tag{S1}$$

$$\mathbf{s}^{(\ell)}[t] = \text{Clip}(\text{Round}(\mathbf{v}^{(\ell)}[t]), -\frac{D}{2} - 1, \frac{D}{2}), \tag{S2}$$

$$\mathbf{h}^{(\ell)}[t] = \mathbf{v}^{(\ell)}[t] \cdot (1 - \mathbf{s}^{(\ell)}[t]) + \mathbf{v}_{reset} \cdot \mathbf{s}^{(\ell)}[t]. \tag{S3}$$

The membrane potential $\mathbf{v}^{(\ell)}[t]$ is computed by summing the previous hidden state $\mathbf{h}^{(\ell)}[t-1]$ and the transformed input signal $\mathbf{x}^{(\ell-1)}[t]$ through the weights $\mathbf{w}^{(\ell)}$. The resulting potential is then rounded and clipped into a valid integer range $\left[-\frac{D}{2} - 1, \frac{D}{2}\right]$ to produce the spike signal $\mathbf{s}^{(\ell)}[t]$. Depending on whether a spike occurs, the hidden state $\mathbf{h}^{(\ell)}[t]$ is either retained or reset to $v_{\text{reset}}$ at the spiking positions. To ensure $\frac{D}{2}$ is an integer, we define it as $\left\lfloor \frac{D}{2} \right\rfloor$ by applying the floor operation.

**Step Two**  To generate bidirectionally encoded spike values of $-1/0/1$, the integer spike counts are mapped by extending the virtual time step from $T$ to $T \times \max\left(\left|-\frac{D}{2} - 1\right|, \left|\frac{D}{2}\right|\right)$. In this process, the input spike signal $s^{(\ell)}[t]$ is expanded into a spike train $\{s^{(\ell)}[t, d]\}_d^{\frac{D}{2}+1}$, effectively distributing the original integer value into a temporally spread train of bidirectional spikes. The corresponding computation is defined as:

$$\mathbf{v}^{(\ell+1)}[t] = \mathbf{h}^{(\ell+1)}[t-1] + \sum_d^{\frac{D}{2}+1} \left(\mathbf{w}^{(\ell+1)}\mathbf{s}^{(\ell)}[t, d]\right). \tag{S4}$$

The membrane potential in the $(\ell+1)$-th layer is then updated based on both the previous hidden state $\mathbf{h}^{(\ell+1)}[t-1]$ and the weighted sum of binary spikes across all virtual steps, using the weight matrix $\mathbf{w}^{(\ell+1)}$.

### A.2  SPIKE FIRING COUNT

Under bidirectional spike encoding, neural outputs take values in $\{-1, 0, +1\}$, where both nonzero components are interpreted as distinct forms of activation. Regardless of direction, all nonzero spikes

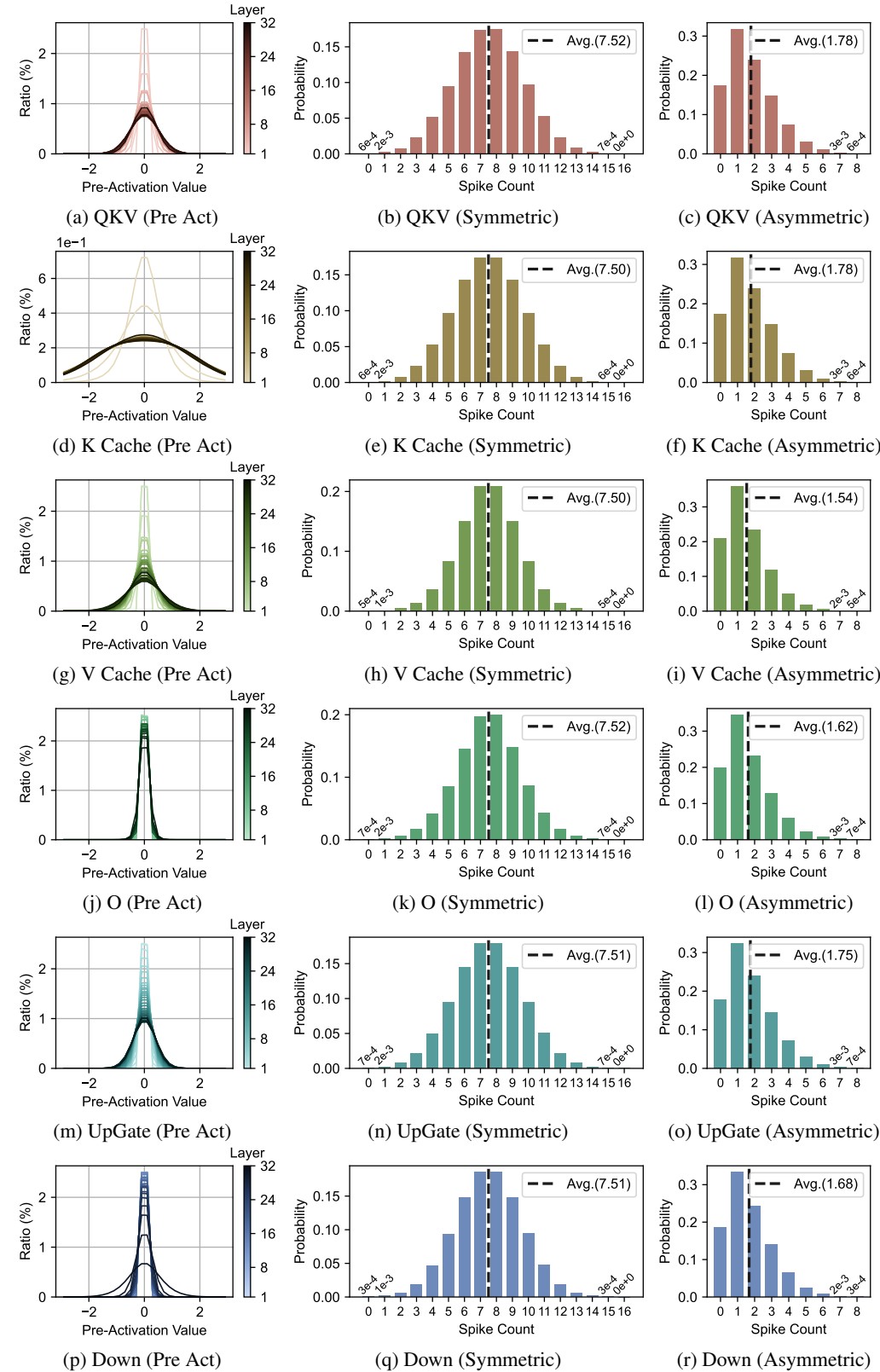

Figure S1: Significant reduction in spike count after symmetric quantization and bidirectional encoding.

are considered functionally equivalent activation events. Accordingly, the total spike activity at layer $\ell$ is computed as:

$$k^{(\ell)} = \sum_t^T \sum_d^{\frac{D}{2}+1} \left| \mathbf{s}^{(\ell)}[t,d] \right|. \tag{S5}$$

Here, $k^{(\ell)}$ denotes the aggregate number of spikes, irrespective of polarity, offering a unified measure of temporal activation under the bidirectional scheme.

## B    VISUALIZATION OF SYMMETRIC VS. ASYMMETRIC SPIKE QUANTIZATION

In Fig. S1, we visualize the pre-activation values and the corresponding spike quantization results for the activations and KV cache components of LLaMA-2 7B. The spike count per time window reflects the overall firing rate. It is evident that symmetric quantization with bidirectional encoding (using -1/0/1 spikes) leads to significantly sparser activity compared to asymmetric quantization (using 0/1 spikes). This highlights the efficiency benefits of symmetric spike quantization and bidirectional encoding in reducing neural activity.

## C    REPLACING MATRIX MULTIPLICATION WITH SPARSE ADDITIONS

Fig. S2 illustrates how the matrix multiplication operator can be transformed into a sparse addition process through spike-based encoding. On the left, a continuous activation vector is first quantized in two steps and expanded into 0/1 spike trains over multiple time steps. During the weight computation, additions are performed only at positions where spikes occur. These spike positions are used to index the corresponding columns of the weight matrix, and the associated weights are summed. This mechanism bypasses traditional dense matrix multiplication, replacing it with sparse, event-driven additions. As a result, it significantly improves inference efficiency and reduces computational energy consumption.

To formalize this computation, we present two theorems showing how dense matrix multiplication can be replaced with sparse additions based on spike events.

**Theorem 2** (Substituting Matrix Multiplication with Sparse Addition from 0/1 Spikes). *Given an input spike train $\mathbf{X} \in \{0,1\}^n$, the dense matrix multiplication $\mathbf{Y} = \mathbf{W}\mathbf{X}$, where $\mathbf{W} \in \mathbb{R}^{m \times n}$, is equivalent to a sparse addition over selected columns of $\mathbf{W}$:*

$$\mathbf{Y} = \sum_{i \in \mathcal{I}} \mathbf{W}_{:,i}, \quad \text{where } \mathcal{I} = \{i \mid X_i = 1\}.$$

*Proof.* Since each element of the input vector $\mathbf{X}$ is binary ($X_i \in \{0,1\}$), the multiplication $W_{j,i} \cdot X_i$ simplifies to:

$$W_{j,i} \cdot X_i = \begin{cases} W_{j,i}, & \text{if } X_i = 1 \\ 0, & \text{if } X_i = 0 \end{cases}$$

Therefore, the matrix-vector product $\mathbf{Y} = \mathbf{W}\mathbf{X}$ can be rewritten as a summation over the columns of $\mathbf{W}$ corresponding to indices $i$ where $X_i = 1$. This eliminates all multiplications with 0, resulting in sparse addition:

$$\mathbf{Y} = \sum_{i \in \mathcal{I}} \mathbf{W}_{:,i}.$$

This shows that when $\mathbf{X}$ is a 0/1 spike vector, the dense computation degenerates into a sparse event-driven process, where only active spikes contribute to the output. □

**Theorem 3** (Substituting Matrix Multiplication with Sparse Addition from -1/0/1 Spikes). *Given an input vector $\mathbf{X} \in \{-1,0,1\}^n$, the matrix multiplication $\mathbf{Y} = \mathbf{W}\mathbf{X}$ can be equivalently computed as a sparse accumulation over selected columns of $\mathbf{W}$, weighted by the sign of spike events:*

$$\mathbf{Y} = \sum_{i \in \mathcal{I}_+} \mathbf{W}_{:,i} + \sum_{i \in \mathcal{I}_-} (-\mathbf{W}_{:,i}), \quad \text{where } \mathcal{I}_+ = \{i \mid X_i = 1\}, \mathcal{I}_- = \{i \mid X_i = -1\}.$$

*Proof.* Each nonzero element in $\mathbf{X}$ represents an event-triggered spike at index $i$, and contributes to the output according to:

$$
\tilde{\mathbf{W}}_{:,i} = \begin{cases} \mathbf{W}_{:,i}, & \text{if } X_i = 1 \\ -\mathbf{W}_{:,i}, & \text{if } X_i = -1 \\ 0, & \text{if } X_i = 0 \end{cases}
$$

Thus, instead of computing $\mathbf{WX}$ through dense multiply-accumulate, we perform sparse selection and signed accumulation over active spike positions:

$$
\mathbf{Y} = \sum_{i \in \mathcal{I}_+} \mathbf{W}_{:,i} + \sum_{i \in \mathcal{I}_-} (-\mathbf{W}_{:,i}).
$$

This sparse formulation eliminates multiplications and directly reflects the event-driven nature of bidirectional spike encoding, where each spike corresponds to a column-wise inclusion or exclusion in the final output. $\square$

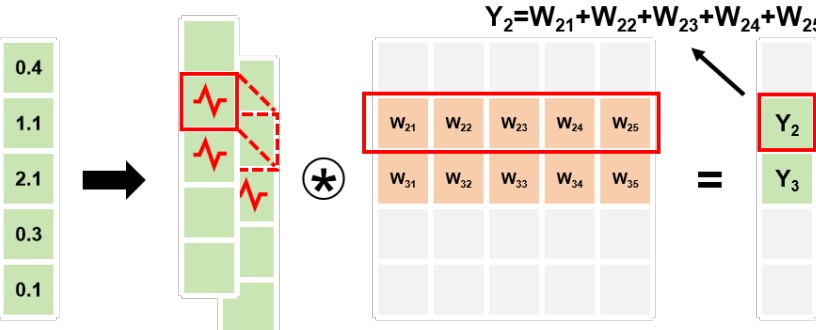

Figure S2: Replacing dense matrix multiplication with sparse addition via spike encoding.

# D ROTATIONAL SPARSE TRAINING

## D.1 TRAINING SETUP

Following the setup in *ReLU Strikes Back* citemirzadehrelu, we fine-tune the pre-trained LLaMA series pre-trained models on the RefinedWeb dataset (Penedo et al., 2023) to evaluate the performance of SDLLM under the membrane potential clipping method. We train on 8 A800 GPUs with approximately 10 million tokens and use the AdamW optimizer with a fixed learning rate of $1.5 \times 10^{-5}$. To improve training efficiency and reduce memory consumption, we adopt the ZeRO Stage 2 optimization strategy (Rajbhandari et al., 2020) provided by DeepSpeed for distributed management of optimizer states and gradients.

## D.2 TRAINING STRATEGY

As discussed in the *Joint Sparsity and Rotation Matrices* subsection of Section 4.4, we adopt a rotational sparse training strategy to enhance quantization performance and activation sparsity during training. Specifically, during training, as illustrated in Fig. S3, we apply an orthogonal rotation matrix $Q$ only to the linear operators whose outputs are involved in sparsification, i.e., those followed by the Quantile-Shifted ReLU activation function. This transformation improves the uniformity of feature distributions and facilitates effective sparsity learning. For operators not participating in sparsification, no rotation is applied during training, thereby avoiding unnecessary computational overhead. During inference, however, we apply the rotation matrix $Q$ uniformly to all linear operators and use the rotated weights $Q^T W$ to ensure compatibility across both sparse and non-sparse computation paths. This strategy strikes a balance between training efficiency and inference consistency, demonstrating the practicality and generalizability of rotational sparse training.

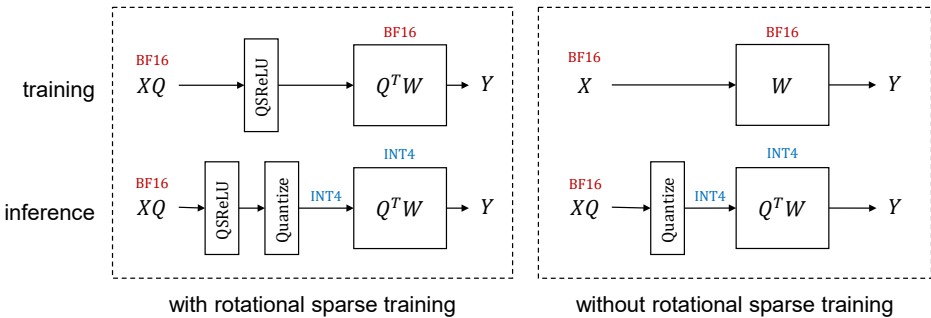

Figure S3: Implementation of rotational sparse training for enhancing spike sparsity.

# E DETAILS OF OPERATIONS AND ENERGY CONSUMPTION

## E.1 ACEs

For $a \in [0, 2^p)$ and $b \in [0, 2^q)$, their binary expansions can be written as $a = \sum_{i=0}^{p-1} a_i 2^i$, $b = \sum_{j=0}^{q-1} b_j 2^j$, $a_i, b_j \in \{0, 1\}$. Then the product $a \cdot b$ can be computed bitwise as Xing et al. (2024a):

$$a \cdot b = \sum_{i=0}^{p-1} \sum_{j=0}^{q-1} 2^{i+j} \operatorname{PopCount}(a_i \& b_j).$$

Here, each $\operatorname{PopCount}(a_i \& b_j)$ corresponds to a multiply–accumulate operation (MAC). Therefore, the total number of operations required to compute $a \cdot b$ is proportional to $p \times q$ MACs, which is exactly the definition of the arithmetic computation effort (ACEs) metric Zhang et al. (2022b).

For $a \in \{0, 1\}$ and $b \in [0, 2^q)$, the binary expansion of $b$ is $b = \sum_{j=0}^{q-1} b_j 2^j$, $b_j \in \{0, 1\}$. Then the product $a \cdot b$ can be expressed as:

$$a \cdot b = \sum_{j=0}^{q-1} 2^j \operatorname{PopCount}(a \& b_j).$$

If $a \in \{-1, 0\}$, the sign bit of $a$ should be separated and does not participate in the bitwise multiplication; in this case, the product can be written as

$$a \cdot b = \operatorname{sign}(a) \cdot \left( \sum_{j=0}^{q-1} 2^j \operatorname{PopCount}(|a| \& b_j) \right),$$

where $\operatorname{sign}(a) \in \{-1, 0\}$ and $|a| \in \{0, 1\}$.

For clarity of comparison between ANN and SNN, we consider the time step of ANN as $T = 1$, so the ACEs metric can be rewritten as ACEs $\times T$.

## E.2 FLOPs

We refer to the FLOPs calculation method for $q$-bit operations from the Q-DETR paper. For 2-bit, 3-bit, and 4-bit operations, the FLOPs for 2-bit operations is $\frac{1}{32}$ of the 32-bit FLOPs, for 3-bit operations it's $\frac{1}{16}$, and for 4-bit operations it's $\frac{1}{8}$ Xu et al. (2023b); Liu et al. (2020), since the current CPU can parallelize bitwise XNOR and popcount operations.

For 4-bit $\times$ 1.5-bit operations, we calculate the FLOPs as 4-bit $\times$ 2-bit, which corresponds to 2 $\times$ 1 operations of 2-bit $\times$ 2-bit FLOPs. Similarly, for 6-bit $\times$ 1.5-bit operations, we calculate the FLOPs as 6-bit $\times$ 2-bit, corresponding to 3 $\times$ 1 operations of 2-bit $\times$ 2-bit FLOPs. For 4-bit $\times$ 4-bit operations, which corresponds to 2 $\times$ 2 operations of 2-bit $\times$ 2-bit FLOPs. For 6-bit $\times$ 6-bit operations, corresponding to 3 $\times$ 3 operations of 2-bit $\times$ 2-bit FLOPs. For ANN with a time step $T$, the FLOPs is written as FLOPs $\times T$.

Table S1: Evaluation of Zero-shot QA (↑) results of LLaMA2-7B and 13B under QLLM settings.

| Method | Spike | W Bit | A Bit | $T = T_D \times R$ | $Range_{(+Z_0)}$ | PIQA | ARC-e | ARC-c | BoolQ | HellaS | WinoG | Avg. | ACEs | Flops (T) | Power (J) |
|---|---|---|---|---|---|---|---|---|---|---|---|---|---|---|---|
| LLAMA-2-7B | ✗ | - | - | - | - | 76.88 | 53.54 | 40.53 | 71.13 | 72.96 | 67.25 | 63.72 | 1× | 6.91 | 33.84 |
| SmoothQuant | ✗ | 6 | 6 | - | $0 \sim 2^6$ | 75.57 | 53.62 | 39.93 | 69.54 | 71.76 | 66.14 | 62.76 | 0.141× | 1.94 | 9.52 |
| OS+ | ✗ | 6 | 6 | - | $0 \sim 2^6$ | 76.22 | 52.74 | 40.70 | - | 71.89 | 65.19 | 61.35 | 0.141× | 1.94 | 9.52 |
| OmniQuant | ✗ | 6 | 6 | - | $0 \sim 2^6$ | 76.55 | 53.83 | 40.96 | 68.75 | 55.89 | 65.59 | 60.26 | 0.141× | 1.94 | 9.52 |
| QLLM | ✗ | 6 | 6 | - | $0 \sim 2^6$ | 77.48 | 52.99 | 39.33 | - | 71.38 | 65.98 | 61.43 | 0.141× | 1.94 | 9.52 |
| DuQuant | ✗ | 6 | 6 | - | $0 \sim 2^6$ | 76.99 | 52.99 | 40.87 | 70.40 | 72.49 | 67.32 | 63.51 | 0.141× | 1.94 | 9.52 |
| SDLLM | ✓ | 6 | 1.5 | $7.1_{=32\times0.222}$ | $0 \sim 2^6$ | 76.99 | 53.75 | 41.04 | 70.64 | 72.84 | 67.25 | **63.75** | 0.333× | 4.60 | **4.14** |
| LLAMA2-13B | ✗ | - | - | - | - | 79.05 | 57.91 | 44.20 | 69.02 | 76.60 | 69.69 | 66.08 | 1× | 13.42 | 65.77 |
| SmoothQuant | ✗ | 6 | 6 | - | $0 \sim 2^6$ | 78.29 | 57.41 | 43.86 | 69.50 | 75.02 | 66.93 | 65.17 | 0.141× | 3.77 | 18.49 |
| OS+ | ✗ | 6 | 6 | - | $0 \sim 2^6$ | 78.29 | 59.13 | 43.34 | - | 75.37 | 67.56 | 64.74 | 0.141× | 3.77 | 18.49 |
| OmniQuant | ✗ | 6 | 6 | - | $0 \sim 2^6$ | 78.24 | 57.58 | 43.86 | 71.10 | 75.52 | 68.35 | 65.78 | 0.141× | 3.77 | 18.49 |
| AffineQuant | ✗ | 6 | 6 | - | $0 \sim 2^6$ | 78.35 | 57.58 | 43.34 | 66.73 | 74.71 | 68.59 | 64.88 | 0.141× | 3.77 | 18.49 |
| QLLM | ✗ | 6 | 6 | - | $0 \sim 2^6$ | 78.78 | 58.29 | 43.77 | - | 75.10 | 68.43 | 64.87 | 0.141× | 3.77 | 18.49 |
| DuQuant | ✗ | 6 | 6 | - | $0 \sim 2^6$ | 78.62 | 56.94 | 43.43 | 68.35 | 76.19 | 69.22 | 65.46 | 0.141× | 3.77 | 18.49 |
| SDLLM | ✓ | 6 | 1.5 | $6.9_{=32\times0.217}$ | $0 \sim 2^6$ | 79.05 | 57.66 | 44.20 | 67.83 | 76.42 | 69.93 | **65.85** | 0.326× | 8.74 | **7.86** |

Table S2: Evaluation of Zero-shot QA (↑) results of LLaMA3-8B under DuQuant settings.

| Method | Spike | W Bit | A Bit | $T = T_D \times R$ | $Range_{(+Z_0)}$ | PIQA | ARC-e | ARC-c | BoolQ | HellaS | WinoG | Avg. | ACEs | Flops (T) | Power (J) |
|---|---|---|---|---|---|---|---|---|---|---|---|---|---|---|---|
| LLaMA3-8B | ✗ | - | - | - | - | 80.85 | 77.78 | 53.41 | 81.28 | 79.16 | 72.84 | 74.22 | 1× | 7.97 | 39.06 |
| SmoothQuant | ✗ | 6 | 6 | - | $0 \sim 2^6$ | 78.94 | 75.88 | 49.49 | 77.58 | 77.39 | 70.80 | 71.68 | 0.141× | 2.24 | 10.98 |
| OmniQuant | ✗ | 6 | 6 | - | $0 \sim 2^6$ | 78.90 | 73.95 | 47.35 | 74.95 | 76.77 | 70.56 | 70.41 | 0.141× | 2.24 | 10.98 |
| AffineQuant | ✗ | 6 | 6 | - | $0 \sim 2^6$ | 78.73 | 73.32 | 46.08 | 74.59 | 77.08 | 70.88 | 70.11 | 0.141× | 2.24 | 10.98 |
| DuQuant | ✗ | 6 | 6 | - | $0 \sim 2^6$ | 80.20 | 77.27 | 52.05 | 80.12 | 79.14 | 72.77 | 73.59 | 0.141× | 2.24 | 10.98 |
| SDLLM | ✓ | 6 | 1.5 | $6.82_{=32\times0.213}$ | $0 \sim 2^6$ | 80.20 | 77.23 | 52.22 | 82.05 | 79.01 | 73.56 | **74.04** | 0.320× | 5.10 | **4.59** |

Table S3: Ablation study of SDLLM on Zero-shot QA (↑) results of LLaMA2-13B.

| Method | Spike | W Bit | A Bit | $T = T_D \times R$ | $Range_{(+Z_0)}$ | PIQA | ARC-e | ARC-c | BoolQ | HellaS | WinoG | Avg. | ACEs | Flops (T) | Power (J) |
|---|---|---|---|---|---|---|---|---|---|---|---|---|---|---|---|
| LLaMA-2-13B | ✗ | - | - | - | - | 80.63 | 77.48 | 49.23 | 80.73 | 79.37 | 71.74 | 80.69 | 1× | 13.42 | 65.77 |
| SDLLM | ✓ | 4 | 1.5 | $1.67_{=8\times0.209}$ | $0 \sim 2^4$ | 78.51 | 74.12 | 46.16 | 78.26 | 76.36 | 69.85 | 70.54 | 0.052× | 1.40 | 1.26 |
| SDLLM | ✓ | 4 | 1.5 | $1.73_{=8.3\times0.209}$ | $0 \sim 2^4\ 94\% / 0 \sim 2^5\ 6\%$ | 79.33 | 73.99 | 47.70 | 77.09 | 76.94 | 69.85 | 70.82 | 0.054× | 1.45 | 1.31 |
| SDLLM | ✓ | 4 | 1.5 | $3.44_{=16\times0.215}$ | $0 \sim 2^5$ | 80.25 | 76.77 | 49.40 | 77.49 | 77.92 | 69.77 | 71.93 | 0.108× | 2.89 | 2.60 |
| SDLLM$_{step1}$ | ✓ | 4 | 4 | 1 | $0 \sim 2^4$ | 78.51 | 74.12 | 46.16 | 78.26 | 76.36 | 69.85 | 70.54 | 0.063× | 1.68 | 8.22 |
| SDLLM | ✓ | 4 | 1.5 | $1.67_{=8\times0.209}$ | $0 \sim 2^4$ | 78.51 | 74.12 | 46.16 | 78.26 | 76.36 | 69.85 | 70.54 | 0.052× | 1.40 | 1.26 |
| SDLLM | ✓ | 4 | 1 | $7.5_{=15\times0.500}$ | $0 \sim 2^4$ | 78.51 | 74.12 | 46.16 | 78.26 | 76.36 | 69.85 | 70.54 | 0.234× | 6.29 | 5.66 |
| SDLLM$_{step1}$ | ✓ | 6 | 6 | 1 | $0 \sim 2^6$ | 80.25 | 76.30 | 48.46 | 80.06 | 79.12 | 72.14 | 72.72 | 0.141× | 3.77 | 18.49 |
| SDLLM | ✓ | 6 | 1.5 | $6.9_{=32\times0.217}$ | $0 \sim 2^6$ | 80.25 | 76.30 | 48.46 | 80.06 | 79.12 | 72.14 | 72.72 | 0.326× | 8.74 | 7.86 |
| SDLLM | ✓ | 6 | 1 | $31.5_{=63\times0.500}$ | $0 \sim 2^6$ | 80.25 | 76.30 | 48.46 | 80.06 | 79.12 | 72.14 | 72.72 | 0.738× | 39.63 | 35.67 |

### E.3 POWER

We refer to the energy consumption metrics from works like SFA, with a 32-bit floating-point implementation in 45nm technology, where $E_{MAC} = 4.6$ pJ and $E_{AC} = 0.9$ pJ Yao et al. (2025; 2024a); Luo et al. (2024). Similarly, for cases with time step $T$, we set $T = 1$ for ANN, then $E_{MAC}$ becomes $E_{MAC} \times T$ and $E_{AC}$ becomes $E_{AC} \times T$.

## F   MORE RESULTS

**Zero-shot QA Results for 6-bit LLaMA Family**   Tab. S1 and S2 present a comparison of zero-shot QA performance under the W6A6 configuration between SDLLM and several mainstream quantization methods, including SmoothQuant, OmniQuant, AffineQuant, and DuQuant, on LLaMA-2 (7B and 13B) and LLaMA-3 (8B). The results show that even under the higher-precision W6A6 setting, SDLLM achieves approximately 2× lower power consumption compared to traditional ANN

Table S4: Comparison of PPL ($\downarrow$) metrics on Wikitext2 and C4 for LLaMA2-7B and LLaMA2-13B between SDLLM and QuaRot.

| Method | Spike | W Bit | A Bit | A $T_{=T_D \times R}$ | A $Range_{(+Z_0)}$ | Wiki | C4 | ACEs | Flops (T) | Power (J) |
|---|---|---|---|---|---|---|---|---|---|---|
| LLaMA2-7B | ✗ | - | - | - | - | 5.47 | 7.26 | 1× | 6.91 | 33.84 |
| SmoothQuant | ✗ | 4 | 4 | - | $0 \sim 2^4$ | 83.12 | 77.27 | 0.063× | 0.86 | 4.23 |
| OmniQuant | ✗ | 4 | 4 | - | $0 \sim 2^4$ | 14.26 | 18.02 | 0.063× | 0.86 | 4.23 |
| AfineQuant | ✗ | 4 | 4 | - | $0 \sim 2^4$ | 12.69 | 15.76 | 0.063× | 0.86 | 4.23 |
| QLLM | ✗ | 4 | 4 | - | $0 \sim 2^4$ | 11.45 | 13.26 | 0.063× | 0.86 | 4.23 |
| Atom | ✗ | 4 | 4 | - | $0 \sim 2^4$ | 8.40 | 10.96 | 0.063× | 0.86 | 4.23 |
| QuaRot-RTN | ✗ | 4 | 4 | - | $0 \sim 2^4$ | 8.73 | 12.27 | 0.063× | 0.86 | 4.23 |
| SDLLM-RTN | ✓ | 4 | 1.5 | $1.73_{=8 \times 0.216}$ | $0 \sim 2^4$ | 6.41 | 8.58 | 0.054× | 0.75 | 0.67 |
| SDLLM-RTN | ✓ | 4 | 1.5 | $3.54_{=16 \times 0.221}$ | $0 \sim 2^5$ | 5.95 | 7.93 | 0.111× | 1.53 | 1.37 |
| LLaMA2-13B | ✗ | - | - | - | - | 4.88 | 6.73 | 1× | 13.42 | 65.77 |
| SmoothQuant | ✗ | 4 | 4 | - | $0 \sim 2^4$ | 35.88 | 43.19 | 0.063× | 1.68 | 8.22 |
| OmniQuant | ✗ | 4 | 4 | - | $0 \sim 2^4$ | 12.30 | 14.55 | 0.063× | 1.68 | 8.22 |
| AfineQuant | ✗ | 4 | 4 | - | $0 \sim 2^4$ | 11.75 | 13.97 | 0.063× | 1.68 | 8.22 |
| QLLM | ✗ | 4 | 4 | - | $0 \sim 2^4$ | 9.09 | 11.13 | 0.063× | 1.68 | 8.22 |
| Atom | ✗ | 4 | 4 | - | $0 \sim 2^4$ | 6.96 | 9.12 | 0.063× | 1.68 | 8.22 |
| QuaRot-RTN | ✗ | 4 | 4 | - | $0 \sim 2^4$ | 6.31 | 9.02 | 0.063× | 1.68 | 8.22 |
| SDLLM-RTN | ✓ | 4 | 1.5 | $1.67_{=8 \times 0.209}$ | $0 \sim 2^4$ | 5.49 | 7.61 | 0.052× | 1.40 | 1.26 |
| SDLLM-RTN | ✓ | 4 | 1.5 | $3.44_{=16 \times 0.215}$ | $0 \sim 2^5$ | 5.18 | 7.15 | 0.108× | 2.89 | 2.60 |

Table S5: Evaluation of Zero-shot QA ($\uparrow$) results of Qwen2.5-14B.

| Method | Spike | W Bit | A Bit | A $T_{=T_D \times R}$ | A $Range_{(+Z_0)}$ | PIQA | ARC-e | ARC-c | BoolQ | HellaS | WinoG | Avg. | ACEs | Flops (T) | Power (J) |
|---|---|---|---|---|---|---|---|---|---|---|---|---|---|---|---|
| Qwen2.5-14B | ✗ | - | - | - | - | 82.10 | 79.12 | 58.87 | 85.26 | 82.91 | 75.30 | 77.26 | 1× | 13.53 | 62.23 |
| RTN | ✗ | 4 | 4 | - | $0 \sim 2^4$ | 51.31 | 32.91 | 24.32 | 50.40 | 29.29 | 47.91 | 39.35 | 0.063× | 1.69 | 7.78 |
| GPTQ | ✗ | 4 | 4 | - | $0 \sim 2^4$ | 51.80 | 26.64 | 23.63 | 41.13 | 26.27 | 49.17 | 36.73 | 0.063× | 1.69 | 7.78 |
| SmoothQuant | ✗ | 4 | 4 | - | $0 \sim 2^4$ | 51.20 | 26.09 | 26.54 | 41.13 | 26.27 | 49.17 | 36.73 | 0.063× | 1.69 | 7.78 |
| SDLLM | ✓ | 4 | 1.5 | $1.7_{=8 \times 0.212}$ | $0 \sim 2^4$ | 79.00 | 77.82 | 52.13 | 80.00 | 78.28 | 67.80 | **72.51** | **0.053×** | **1.43** | **1.29** |
| SDLLM | ✓ | 4 | 1.5 | $3.4_{=16 \times 0.213}$ | $0 \sim 2^5$ | 81.28 | 80.43 | 55.29 | 82.64 | 81.41 | 74.82 | **76.15** | **0.107×** | 2.88 | **2.59** |
| RTN | ✗ | 6 | 6 | - | $0 \sim 2^6$ | 80.41 | 81.40 | 56.57 | 84.19 | 81.52 | 71.27 | 75.89 | 0.141× | 3.81 | 17.50 |
| GPTQ | ✗ | 6 | 6 | - | $0 \sim 2^6$ | 79.71 | 76.85 | 52.82 | 80.24 | 80.11 | 70.17 | 73.32 | 0.141× | 3.81 | 17.50 |
| SmoothQuant | ✗ | 6 | 6 | - | $0 \sim 2^6$ | 79.33 | 78.96 | 55.03 | 80.89 | 79.12 | 68.67 | 73.66 | 0.141× | 3.81 | 17.50 |
| SDLLM | ✓ | 6 | 1.5 | $6.9_{=32 \times 0.215}$ | $0 \sim 2^6$ | 82.48 | 79.04 | 57.76 | 84.83 | 82.85 | 75.22 | **77.03** | 0.323× | 8.73 | **7.85** |

Table S6: Evaluation of PPL ($\downarrow$) results of Qwen2.5-14B.

| Method | Spike | W Bit | A Bit | A $T_{=T_D \times R}$ | A $Range_{(+Z_0)}$ | Wiki | C4 | ACEs | Flops (T) | Power (J) |
|---|---|---|---|---|---|---|---|---|---|---|
| Qwen2.5-14B | ✗ | - | - | - | - | 5.29 | 10.35 | 1× | 13.53 | 62.23 |
| RTN | ✗ | 4 | 4 | - | $0 \sim 2^4$ | 2e3 | 2e3 | 0.063× | 1.69 | 7.78 |
| GPTQ | ✗ | 4 | 4 | - | $0 \sim 2^4$ | 6e3 | 4e3 | 0.063× | 1.69 | 7.78 |
| SmoothQuant | ✗ | 4 | 4 | - | $0 \sim 2^4$ | 2e4 | 2e4 | 0.063× | 1.69 | 7.78 |
| SDLLM | ✓ | 4 | 1.5 | $1.70_{=8 \times 0.212}$ | $0 \sim 2^4$ | 8.19 | 16.12 | 0.053× | 1.43 | 1.29 |
| SDLLM | ✓ | 4 | 1.5 | $3.41_{=16 \times 0.213}$ | $0 \sim 2^5$ | 6.13 | 11.14 | 0.107× | 2.88 | 2.59 |

quantization, while achieving SOTA performance. This demonstrates the potential of spike-based sparse inference at higher bit-widths.

**Zero-shot QA and PPL Results for Qwen2.5-14B**  We validated our proposed method on the newer LLaMA model Qwen2.5-14B. According to the results in Tab. S5 and S6, SDLLM continues to perform excellently in Zero-shot QA and PPL tasks, while significantly reducing the number of operations and energy consumption. For example, in the case of W4A4, compared to quantization methods, SDLLM reduces the number of operations by 1.2× and energy consumption by 6×.

Table S7: Evaluation of more complex language tasks (↑) on the LLaMA family: reading Comprehension (SQuAD), world Knowledge (TriviaQA), and math (GSM8K)

| Method | Spike | W Bit | A Bit | $T_{=T_D \times R}$ | $Range_{(+Z_0)}$ | GSM8K Str. | Flex. | SQuAD EM | F1 | HA-EM | HA-F1 | NA-F1 | TriviaQA EM | ACEs | Flops (T) | Power (J) |
|---|---|---|---|---|---|---|---|---|---|---|---|---|---|---|---|---|
| Llama-2-7B | ✗ | - | - | - | - | 15.30 | 15.30 | 16.77 | 24.21 | 24.68 | 39.37 | 8.66 | 64.14 | 1× | 6.91 | 33.84 |
| QuaRot | ✗ | 4 | 4 | - | $0 \sim 2^4$ | 1.97 | 2.73 | 17.04 | 24.76 | 29.01 | 44.26 | 4.77 | 33.20 | 0.063× | 0.86 | 4.23 |
| SDLLM | ✓ | 4 | 1.5 | $1.73_{=8 \times 0.216}$ | $0 \sim 2^4$ | 6.36 | 6.67 | 19.16 | 25.52 | 18.96 | 31.52 | 19.37 | 51.71 | 0.054× | 0.75 | 0.67 |
| SDLLM | ✓ | 4 | 1.5 | $3.54_{=16 \times 0.221}$ | $0 \sim 2^5$ | 10.00 | 10.15 | 13.88 | 22.53 | 20.76 | 37.86 | 6.82 | 58.86 | 0.111× | 1.53 | 1.37 |
| QuaRot | ✗ | 6 | 6 | - | $0 \sim 2^6$ | 13.48 | 13.79 | 16.50 | 24.37 | 22.22 | 37.77 | 10.64 | 62.72 | 0.063× | 1.94 | 9.52 |
| SDLLM | ✓ | 6 | 1.5 | $7.1_{=32 \times 0.222}$ | $0 \sim 2^6$ | 14.85 | 1.39 | 18.05 | 25.12 | 24.82 | 38.77 | 11.12 | 64.05 | 0.333× | 4.60 | 4.14 |
| Llama-2-13B | ✗ | - | - | - | - | 22.73 | 22.88 | 22.77 | 29.67 | 37.19 | 50.82 | 7.98 | 70.45 | 1× | 13.42 | 65.77 |
| QuaRot | ✗ | 4 | 4 | - | $0 \sim 2^4$ | 12.42 | 12.42 | 21.96 | 29.62 | 41.78 | 56.90 | 1.64 | 52.25 | 0.063× | 1.68 | 8.22 |
| SDLLM | ✓ | 4 | 1.5 | $1.67_{=8 \times 0.209}$ | $0 \sim 2^4$ | 15.91 | 16.21 | 21.29 | 28.49 | 37.52 | 51.75 | 4.64 | 62.44 | 0.052× | 1.40 | 1.26 |
| SDLLM | ✓ | 4 | 1.5 | $3.44_{=16 \times 0.215}$ | $0 \sim 2^5$ | 21.06 | 21.36 | 21.15 | 28.82 | 33.07 | 48.21 | 8.94 | 66.45 | 0.108× | 2.89 | 2.60 |
| QuaRot | ✗ | 6 | 6 | - | $0 \sim 2^6$ | 21.82 | 21.97 | 23.85 | 30.42 | 41.25 | 54.24 | 6.00 | 69.14 | 0.063× | 3.77 | 18.49 |
| SDLLM | ✓ | 6 | 1.5 | $6.9_{=32 \times 0.217}$ | $0 \sim 2^6$ | 21.52 | 21.82 | 21.99 | 29.06 | 34.46 | 48.43 | 9.21 | 69.84 | 0.326× | 8.74 | 7.86 |
| Llama-3-8B | ✗ | - | - | - | - | 48.64 | 49.55 | 26.71 | 32.58 | 52.69 | 64.30 | 0.07 | 71.58 | 1× | 7.97 | 39.06 |
| SDLLM | ✓ | 4 | 1.5 | $1.68_{=8 \times 0.210}$ | $0 \sim 2^4$ | 18.80 | 19.41 | 19.65 | 27.43 | 37.66 | 52.59 | 0.05 | 49.31 | 0.053× | 0.84 | 0.75 |
| SDLLM | ✓ | 4 | 1.5 | $3.38_{=16 \times 0.211}$ | $0 \sim 2^5$ | 31.84 | 32.60 | 24.55 | 30.89 | 48.37 | 60.89 | 0.14 | 60.40 | 0.106× | 1.68 | 1.51 |
| SDLLM | ✓ | 6 | 1.5 | $6.82_{=32 \times 0.213}$ | $0 \sim 2^6$ | 46.55 | 47.16 | 28.32 | 34.74 | 54.31 | 66.61 | 0.05 | 70.26 | 0.320× | 5.09 | 4.58 |

Table S8: Spike Firing Details and FLOPs of Linear Layers in LLaMA2-7B

| Model | Layer | Time Complexity | $T$ | $R$ | FLOPs (G) | Power (mJ) |
|---|---|---|---|---|---|---|
| LLaMA2-7B W4A1.5 | k_proj | $ND_h^2$ | 8 | 0.2230 | 1.92 | 1.73 |
| | v_proj | $ND_h^2$ | 8 | 0.2230 | 1.92 | 1.73 |
| | q_proj | $ND_h^2$ | 8 | 0.2230 | 1.92 | 1.73 |
| | out_proj | $ND_h^2$ | 8 | 0.2028 | 1.74 | 1.57 |
| | gate_proj | $ND_h D_i$ | 8 | 0.2189 | 5.05 | 4.55 |
| | up_proj | $ND_h D_i$ | 8 | 0.2189 | 5.05 | 4.55 |
| | down_proj | $ND_h D_i$ | 8 | 0.2096 | 4.84 | 4.36 |
| LLaMA2-7B W4A1.5 | k_proj | $ND_h^2$ | 16 | 0.2257 | 3.88 | 3.49 |
| | v_proj | $ND_h^2$ | 16 | 0.2257 | 3.88 | 3.49 |
| | q_proj | $ND_h^2$ | 16 | 0.2257 | 3.88 | 3.49 |
| | out_proj | $ND_h^2$ | 16 | 0.2192 | 3.77 | 3.39 |
| | gate_proj | $ND_h D_i$ | 16 | 0.2212 | 10.21 | 9.19 |
| | up_proj | $ND_h D_i$ | 16 | 0.2212 | 10.21 | 9.19 |
| | down_proj | $ND_h D_i$ | 16 | 0.2192 | 10.12 | 9.11 |
| LLaMA2-7B W6A1.5 | k_proj | $ND_h^2$ | 32 | 0.2284 | 11.77 | 10.59 |
| | v_proj | $ND_h^2$ | 32 | 0.2284 | 11.77 | 10.59 |
| | q_proj | $ND_h^2$ | 32 | 0.2284 | 11.77 | 10.59 |
| | out_proj | $ND_h^2$ | 32 | 0.2217 | 11.43 | 10.29 |
| | gate_proj | $ND_h D_i$ | 32 | 0.2237 | 30.99 | 27.89 |
| | up_proj | $ND_h D_i$ | 32 | 0.2237 | 30.99 | 27.89 |
| | down_proj | $ND_h D_i$ | 32 | 0.2133 | 29.54 | 26.59 |

**PPL Results for LLaMA Family**   Tab. S4 shows a comparison of PPL between SDLLM and QuaRot under W4A4 quantization precision for the LLaMA2-7B and LLaMA2-13B models. SDLLM significantly outperforms QuaRot, reducing perplexity by 26.6% and 29.9% on the WikiText2 and C4 datasets, respectively (LLaMA2-7B), while also reducing ACEs by 1.17×, FLOPs by 1.15×, and energy consumption by 6.3×. For LLaMA2-13B, SDLLM improves model performance under low-precision quantization, reducing perplexity by 13.0% and 15.7%, while reducing ACEs by 1.21×, FLOPs by 1.2×, and energy consumption by 6.5×.

**More Complex Tasks for LLaMA Family**   In addition to performing well in commonsense reasoning tasks (such as PIQA, ARC-easy, ARC-challenge, HellaSwag and WinoGrande), we further extended our evaluation to more complex language generation tasks, including reading comprehension (BoolQ, SQuAD), world knowledge (TriviaQA), and mathematical problem solving (GSM8K). These tasks assess the model's performance in different domains, particularly those that require higher reasoning abilities and domain knowledge. The results show that SDLLM demonstrates strong adaptability and excellent performance in these complex tasks. Especially under low-precision

quantization (such as the W4A4 configuration), it significantly improves the model's reasoning efficiency, while also showing advantages in reducing energy consumption and computational resources.

## G SPIKE FIRING DETAILS

As mentioned earlier in Section 4.1, the computational cost of non-matrix multiplication operators is several orders of magnitude lower. Therefore, in Tab. S8, we present the spike firing behavior and corresponding FLOPs of the linear layers in the SDLLM based on the LLaMA-2 7B baseline. In all tables, $N$ denotes the train length, and we uniformly set $N = 1024$. In addition, $D_h$ and $D_i$ represent the hidden size and intermediate size, respectively. In addition to applying spiking to the linear layers, we also spiked the KV Cache, similar to how quantization methods process the KV Cache. The spiked KV Cache is directly involved in the computation of spiking attention.

## H HARDWARE STRATEGIES

In hardware implementation, three different design strategies can be considered: Serial, Parallel, and Parallel Reuse, as illustrated in Fig. S5 Among them, Serial is the most fundamental, where each time step is computed sequentially. However, since the effective number of time steps in our setting is usually less than 2, the delay overhead can be neglected. The Parallel strategy allows multiple time steps to be computed simultaneously, thereby eliminating delay, but it requires higher memory and hardware resources. To strike a balance between the two, the Parallel Reuse strategy processes a fixed number of time steps in parallel and reuses the same computation units, thus achieving an optimal trade-off between latency and memory overhead.

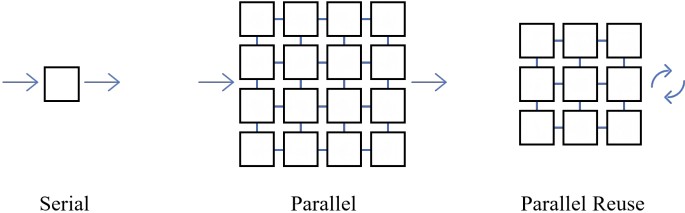

Serial            Parallel            Parallel Reuse

Figure S4: Illustration of three hardware strategies: Serial, Parallel, and Parallel Reuse.

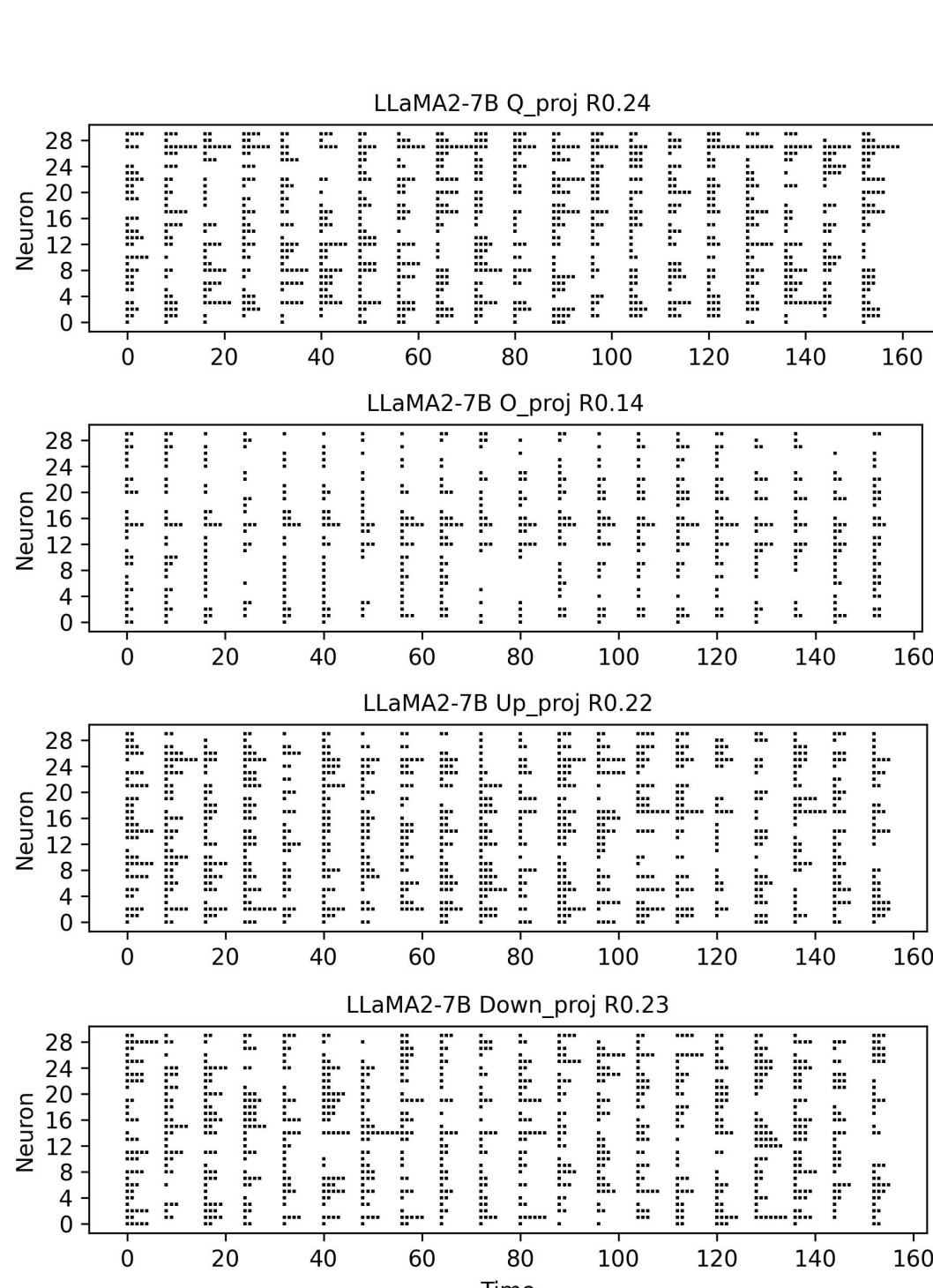

Figure S5: Ternary spike visualization in LLaMA2-7B. Time is token time $\times$ T$_D$.

