# OpenReview forum: "Spike-driven Large Language Model"
_ICLR.cc/2026/Conference — ICLR 2026 Conference Withdrawn Submission_

### Official Review · Reviewer_v6Qd · 2025-10-26

**Soundness:** 2
**Presentation:** 2
**Contribution:** 2
**Rating:** 4
**Confidence:** 4

**Summary:**

This paper propose a spike-driven large language model SDLLM that using sparse additions. A two steps spike generation method is proposed to replace MAC with ACC and a quantile-shifted ReLU is used to increase sparsity.

**Strengths:**

Since there’s almost no spiking LLMs (like spiking LLaMA) related work, this paper explore with adaptation of SSN for those large models.

**Weaknesses:**

Please refer to the Question section.

**Questions:**

1. Line 86: Typo, reads "LLM We".
2. Section 3, Line 149: Should this be "IF" instead of "LIF"? Also, where is the decay function explained?
3. Line 190: Regarding T*D, does this mean you expand a k-bit integer into a (k^2 - 1) spike train? More details should be explained here since this would imply additional storage costs which may eliminate the advantages of replacing MAC with ACC or even consume more energy.
4. Line 206: I checked the code, and it appears to be disorganized. The root README file seems to be from another published paper. After a careful search, I cannot find the approximation for RMSNorm and softmax. It seems the author uses common RMSNorm and softmax implementations (found in quant_utils.py line 459; trainer.py lines 105, 106). If authors declare that their method is purely addition-based (like both in abstract “line15 eliminating matrix multiplications and relying solely on sparse additions “ ), they should at least implement the operations they are replacing (like RMSNorm and softmax) for a fair comparison, instead of ignoring them. Simulating these functions properly would likely result in different, and most likely lower, accuracy than what is reported.
5. Line 252:  The latency of SNN is set as T = T_D*R  which looks unfair to me. There exists additional static power consumption among the whole time window size and we usually wait for the slowest component (T_D) to complete before processing the next layer. Could the authors justify setting the latency in this way?
6. Line 293 : Authors declare they propose a new encoding scheme with ternary spikes but this concept (spikes of –1/0/1) is actually not new, can author compare what’s new compare with previous ternary spike encoding methods? (Eg. “Ternary Spike: Learning Ternary Spikes for Spiking Neural Networks” or “SPIKELLM: SCALING UP SPIKING NEURAL NETWORK TO LARGE LANGUAGE MODELS VIA SALIENCY-BASED SPIKING”)
7. Line 298: Why is this value halved to D/2? More explanation is needed.
8. Line 361: The paper's claim of significant energy savings (up to 13x) appears to be based on an overly simplified and unrealistic energy model. The model only compares the theoretical energy of a single MAC versus a single AC, ignoring the primary drivers of energy consumption in modern systems: data movement and control logic overhead. The proposed method's memory access pattern seems highly inefficient and would likely lead to an increase in data movement and control complexity. This overhead could easily nullify the theoretical computational savings. While it might be acceptable to claim energy benefits against other SNNs using this simplified energy calculation way, the comparison to ANNs is problematic. In abstract authors make a very strong claim: "SDLLM is the first to demonstrate that SNNs outperform quantized ANNs in both performance and energy efficiency." This requires very careful validation. Furthermore, this comparison seems unfair. Setting aside the extra storage costs (mentioned in my point for line 190), this event-driven SNN model appears to read weights for every single accumulation. In a standard ANN, the weights are read once. This implies the SNN's memory access frequency is many times higher than the ANN's. The paper cannot exclusively highlight its computational advantages while ignoring its significant memory access disadvantages.
9. Line 372 and Table 1: The "1.5bit" value should likely be "1.5 * T_D” bits. If you are calculating activation bits this way, it's misleading. By this logic, for a 4-bit QANN, could one not just process it four times at 1-bit each and then claim it's a 1-bit model?

---

### Official Review · Reviewer_ZuMi · 2025-10-27

**Soundness:** 2
**Presentation:** 2
**Contribution:** 2
**Rating:** 2
**Confidence:** 3

**Summary:**

The paper is an attempt to integrate SNNs with LLMs while aiming to improve the performance and efficiency of previous attempts. They proposed SDLLM (Spike-Driven LLM) that eliminates matrix multiplication by taking advantage of SNNs paradigm of sparse addition. They introduced a bidirectional encoding that uses symmetric quantization, along with a membrane potential clipping mechanism to reduce energy consumption without compromising accuracy

**Strengths:**

The authors claim that SDLLM is able to reduce energy consumption by 7.8 times compared to previous spike-based LLMs while preserving the accuracy. A large set of experiments is given backs this.

**Weaknesses:**

The paper seems mainly focus on the encoding and training aspects of its model. Unfortunately, I cannot get a more complete picture of what they are doing. According to Figure 1, QKV are simply multiplied together. But that is not the attention equation. Where is softmax (never mind safe softmax and scaling)? It is also not clear how two spike trains can be "added" correctly to form the equivalent of two matrix multiplication results (never mind the softmax in between).

The SNN model appears to use rate encoding. According to Fig. 1, the "bidirectionally" encoded signal seems to double the window size. If so, not only will this impact latency (tokens per second, an important LLM metric not evaluated in the experiments), but it will also impact energy because despite the sparsity, the circuits must be kept on twice the amount of time.

The last bit above brings me to my final criticism: the energy model is way too optimistic. The biggest problem with LLM is data movement, not computation. Indeed, reducing bitwidths will help, but FLOPS counting does not reveal the whole picture. In fact, my own group's profiling reveals that even at very small bitwidths, data movement energy dominates.

The supplementary file is pretty much unusable. The Makefile has hardwired directories, and even when those are fixed, there are missing files. I gave up trying to repair it.

**Questions:**

1. How are you dealing with softmax and other complex operations such as scaling, dropouts and masking?

2. What if data storage and movement is also considered in your energy model? How will that change things?

3. How does SDLLM compare to ternary model such as BitNet? https://arxiv.org/abs/2402.17764

4. How does it compare to W4A2 SNNs?

**Details Of Ethics Concerns:**

None.

---

### Official Review · Reviewer_PmtU · 2025-10-29

**Soundness:** 3
**Presentation:** 3
**Contribution:** 2
**Rating:** 4
**Confidence:** 4

**Summary:**

The authors in this paper propose a matmul free LLM namely, Spike-driven Large Language Model (SDLLM). The proposed two-step spike quantization strategy addresses outliers in activation values and reduces accuracy loss. The authors evaluate their proposed technique across multiple language modeling and commonsense QA tasks and demonstrate competitive performance and reduced power consumption.

**Strengths:**

1. The paper is well written. The motivation is clear and well founded since efficient LLM is a relevant research domain for the current ML community.

2. The paper achieves competitive results in the domain of SpikingLLMs.

**Weaknesses:**

1. The latency introduced into the model for the two-step quantization process is not quantized in the paper. Furthermore, quantile computation in real-time as well as multiplication with rotational matrix Q, all adds extra computation and latency which are not elaborated properly.
2. The paper is more focused towards quantization than actual bio-plausibility. Is there any bio-plausible interpretation for Eqn. 6?
3. Since, the authors focus if primary on energy-efficiency and the work is highly empirical, I don't think only theoretical estimates for energy efficiency is suffiicient.
4. While the authors note that SpikeLLM relies on 8-bit high activation values, the original SpikeLLM paper also reports support for 1-bit inference. Could the authors please clarify how their proposed method differs from and improves upon SpikeLLM in terms of efficiency?

**Questions:**

Please see weaknesses

Additional Question:

1. Can the authors please explain how T = Td * R is formulated. Is the time required for quantization in Step-1 accounted into the calculation?

2. Also, could the authors please explain Fig 1b. There seems to be lot more time steps after step 2 quantization.

---

### Official Review · Reviewer_L81q · 2025-11-02

**Soundness:** 2
**Presentation:** 3
**Contribution:** 2
**Rating:** 4
**Confidence:** 4

**Summary:**

The paper proposes SDLLM, a spike-driven large language model designed to improve energy efficiency while maintaining high performance. It replaces traditional matrix multiplications with sparse additions through spike-driven computation and introduces a two-stage quantization scheme with symmetric ternary spike encoding to handle the challenges of quantization and activation outliers. The model also employs bidirectional encoding and membrane potential clipping to reduce spike firing rates, further enhancing sparsity and energy savings. Experimental results demonstrate that SDLLM achieves significant energy reductions—up to 7.8 times— and surpasses prior spike-based approaches in accuracy, thereby bridging the gap between neuromorphic computing and modern large language models.

**Strengths:**

- The method significantly reduces power consumption, achieving less than one-third of the energy used by traditional quantization approaches while maintaining competitive accuracy.

- By eliminating matrix multiplications, employing a two-stage quantization process, and using symmetric ternary spike encoding, the approach enhances sparsity and energy savings without substantial accuracy loss.

- The core ideas are described as relatively straightforward and intuitive, which could facilitate understanding and potential adoption.

**Weaknesses:**

- The experiments primarily focus on a limited set of models and tasks, mainly on commonsense QA datasets and a few language models like LLaMA2 variants. Whether the proposed approach will generalize well to other models and diverse NLP tasks remains unclear. In particular, a broader evaluation across multiple models is needed to demonstrate robustness.

- The proposed method adopts techniques such as symmetric (ternary) spike encoding and engineered components like negative spikes, which diverge from traditional, biologically plausible SNNs. The alignment with biological inspiration and practical hardware feasibility, especially on neuromorphic platforms, is yet to be confirmed.

- The aggressive sparsification and quantization might cause information loss or reduce the model’s capacity to handle complex or nuanced tasks. The two-step quantization could obscure fine-grained temporal coding, potentially impacting modeling expressivity.

- While the impressive energy efficiency and accuracy results are encouraging, the incomplete evaluation metrics, such as the absence of perplexity (PPL) comparisons with other quantization methods, are a concern. It is recommended to include experiments that measure actual wall-clock inference time on hardware to validate theoretical efficiency claims.

- The paper relies on some heuristics and manually tuned hyperparameters. The lack of comprehensive sensitivity analyses raises questions about the robustness of the approach under different settings.

**Questions:**

See weakness.

---

### Note · Authors · 2025-11-27

I have read and agree with the venue's withdrawal policy on behalf of myself and my co-authors.